# Hybrid Quality-Based Recommender Systems: A Systematic Literature Review [note 1]

**DOI:** 10.3390/jimaging11010012

**Published:** 2025-01-07

**Authors:** Bihi Sabiri, Amal Khtira, Bouchra El Asri, Maryem Rhanoui

**Affiliations:** 1IMS Team, ADMIR Laboratory, Rabat IT Center, ENSIAS, Mohammed V University in Rabat, Rabat 10130, Morocco; 2LASTIMI Laboratory, EST Salé, Mohammed V University in Rabat, Salé 11060, Morocco; 3Laboratory Health Systemic Process (P2S), UR4129, University Claude Bernard Lyon 1, University of Lyon, 69008 Lyon, France; 4Meridian Team, LYRICA Laboratory, School of Information Sciences, Rabat 10100, Morocco

**Keywords:** hybrid quality-based recommendations, strategy recommender systems, systematic review, big data

## Abstract

As technology develops, consumer behavior and how people search for what they want are constantly evolving. Online shopping has fundamentally changed the e-commerce industry. Although there are more products available than ever before, only a small portion of them are noticed; as a result, a few items gain disproportionate attention. Recommender systems can help to increase the visibility of lesser-known products. Major technology businesses have adopted these technologies as essential offerings, resulting in better user experiences and more sales. As a result, recommender systems have achieved considerable economic, social, and global advancements. Companies are improving their algorithms with hybrid techniques that combine more recommendation methodologies as these systems are a major research focus. This review provides a thorough examination of several hybrid models by combining ideas from the current research and emphasizing their practical uses, strengths, and limits. The review identifies special problems and opportunities for designing and implementing hybrid recommender systems by focusing on the unique aspects of big data, notably volume, velocity, and variety. Adhering to the Cochrane Handbook and the principles developed by Kitchenham and Charters guarantees that the assessment process is transparent and high in quality. The current aim is to conduct a systematic review of several recent developments in the area of hybrid recommender systems. The study covers the state of the art of the relevant research over the last four years regarding four knowledge bases (ACM, Google Scholar, Scopus, and Springer), as well as all Web of Science articles regardless of their date of publication. This study employs ASReview, an open-source application that uses active learning to help academics filter literature efficiently. This study aims to assess the progress achieved in the field of hybrid recommender systems to identify frequently used recommender approaches, explore the technical context, highlight gaps in the existing research, and position our future research in relation to the current studies.

## 1. Introduction

Based on extensive datasets, a recommender system is defined as any system that generates personalized suggestions as an output or has the effect of leading the user to interesting or helpful objects in a broad range of alternative options. In the context of big data, recommender systems are a crucial tool for sharing knowledge and assisting users in finding pertinent content.

### 1.1. Current Landscape of E-Commerce

Propelled by significant technology advancements, the e-commerce industry today is marked by a growing rivalry among platforms competing for user attention [1,2]. A sizable fraction of the worldwide population currently shops online, and the e-commerce industry is only expected to expand more. Correspondingly, consumer behavior is shifting toward web-related activities such as online purchasing and product research. With so much variety, e-commerce enterprises must develop inventive ways to retain or attract customers. In this context, robust recommendation systems are critical for identifying diverse forms of data and understanding consumer preferences. Increasing volumes of data and technological improvements have turned the focus to data analytics as businesses now value the insights and helpful patterns that result from the process. In a typical market, a small percentage of loyal clients account for a considerable portion of the future income. This demonstrates the necessity of retaining high customer value and advancing potential customers up the profit chain. Prospective product and service offerings should effectively appeal to consumer preferences, helping them to progress up the loyalty ladder. Social media play an important role in e-commerce nowadays, and enormous volumes of data regarding individual preferences are frequently publicized. The availability of platforms that use various methodologies indicates how diversified recommendation systems contribute to significant improvements in modern e-commerce. The issues ahead are mostly related to information overload and erroneous suggestions across multiple categories. The current situation requires the use of an effective hybrid recommender system to ensure correct targeting of potential clients.

### 1.2. Hybrid Recommendation Systems: Industry Impacts and Applications

Today, major internet businesses such as Amazon, LinkedIn, Google, Facebook, Netflix, Spotify, Microsoft, eBay, and Airbnb use hybrid recommendation algorithms, which have had a significant impact on the global economy, social sphere, and digital space [3]. The following are specific examples of studies and applications that demonstrate how hybrid recommendation systems are used across several sectors:**E-Commerce:** Amazon’s recommendation engine has a hybrid approach, combining collaborative filtering with content-based filtering. For example, the authors of [1] found that Amazon customizes recommendation systems to reflect customer habits and interests, resulting in a huge 35% boost in sales volume on their online shopping platform [4].**Music Streaming:** Spotify: Spotify combines user listening history (collaborative filtering) with musical elements such as genre, pace, and lyrics (content-based filtering). Spotify’s “Discover Weekly” playlist has successfully leveraged these strategies to increase user engagement and satisfaction, resulting in a personalized experience that retains users. As demonstrated by the authors of [5], after implementing their model, the rate at which consumers began new audiobooks increased by 46%.**Online Video Platforms:** Netflix’s recommendation algorithm uses collaborative filtering, content-based approaches, and contextual considerations (such as time of day and device type). Collaborative filtering detects patterns in user viewing behavior, whereas content-based filtering suggests shows based on genre and theme. The authors of [6] discovered that this hybrid strategy improves the user experience and retention rates by providing timely and relevant recommendations.**YouTube:** YouTube’s recommendation system is fundamental to the platform, and it was meticulously engineered to optimize user engagement and time spent watching videos. The system works by evaluating user interaction data, including watch history, likes, comments, shares, and the amount of time spent on different types of content. It learns from these interactions to create a more detailed picture of each viewer’s preferences. In addition to analyzing user behavior, the program examines content. This includes evaluating metadata such as video titles, descriptions, tags, and more complicated aspects such as the topic, style, and tone of the content. By merging these data, the system detects trends and predicts what viewers are likely to watch and appreciate. The final objective of YouTube’s recommendation algorithm is to present each viewer with a personalized feed that will keep them interested for longer times. By proposing content that closely matches their interests, YouTube is able to boost viewers’ viewing time, which benefits both advertisers and content providers while also guaranteeing that consumers continue to find relevant, engaging videos on the site. This cycle of personalized recommendations not only improves the user experience but also promotes YouTube’s status as a top content platform by encouraging long-term and recurring use.**Travel and Hospitality:** Airbnb’s recommendation system personalizes listing suggestions based on user interests, demographics, and geography.**Social Media:****Facebook** employs a hybrid algorithm for friend suggestions that incorporates user interactions, mutual connections, and demographic information. Backstrom et al. (2011) showed that this method promotes user engagement by fostering more meaningful connections.**LinkedIn’s** job recommendation engine incorporates profile information, user behavior, and collaborative filtering. LinkedIn tailors job suggestions based on user data and behaviors from comparable users, increasing the job-seeking relevance and improving the professional networking experience.

These systems have also made contributions to the issues of information overload, user experience, user decision-making, and business sales [7].

Regardless of the goals of any recommendation technique, hybrid recommender systems (HRSs) combine two or more of them to improve the forecast accuracy [8,9,10]. In this manner, the drawbacks of each method that would arise from using them separately can be somewhat mitigated [11,12]. Since advances in technology have made it possible for individuals and businesses to acquire a vast amount of information on any topic, the field of human resource systems has been becoming increasingly relevant. This tendency can be recognized in several areas, including e-learning systems, digital libraries, navigation services, electronic enterprises, and news and publication suggestions. Hybrid recommender systems integrate several models to reduce the shortcomings of one model with another, lowering the overall disadvantages of using different models and resulting in more credible solutions.

The two primary categories of hybrid recommendation systems are collaborative-filtering-based and content-based. While collaborative-filtering-based systems rely on user activity, content-based systems create suggestions based on the characteristics of the things being recommended. Both categories can be used by hybrid recommender systems to create a more successful recommendation engine [13].

One benefit of hybrid systems is their ability to provide users more individualized recommendations. They can consider a wider range of criteria when making suggestions by merging various models, which can produce more accurate and pertinent findings. However, because hybrid systems sometimes combine multiple recommendation systems, they can be more complex and challenging to analyze.

In technical terms, all recommendation systems generate suggestions using various methodologies, such as collaborative filtering (CF), content-based filtering (CBF), knowledge-based filtering (KBF), demographic filtering (DF), and others. Let us consider the specifics of these strategies.

**Content-Based Filtering:** The CBF approach is based on the notion that people who have previously appreciated products with certain characteristics would continue to enjoy similar items in the future. It examines item features to match them to user profiles and provide suggestions. This strategy uses content representation and comparison techniques from information retrieval, as well as classification algorithms from machine learning, to represent those items previously rated by the user and compare them to other items to propose comparable items [3,14,15,16].**Collaborative Filtering:** The CF approach works on the notion that people who had similar preferences in the past would have similar ones in the future. The most significant part of collaborative filtering is determining whether the user’s preferences match those of other users [17]. It entails people working together to help each other filter information by documenting their emotions regarding the things they encounter [3]. To find similarities in taste among groups of people, CF uses ratings or user-generated comments. The commonalities between users are then used to produce recommendations [2]. However, CF recommenders encounter difficulties such as the cold-start problem (for new users or goods) and the “gray sheep” problem (users who do not fall into any single taste cluster) [3,14,18,19].**Utility-Based Filtering:** UBF is a recommendation approach that provides personalized recommendations to users by calculating the utility of each item for the user. However, a key challenge in this category lies in determining the utility value for each individual user [20,21].**Contextual Filtering:** This system considers contextual information such as time, location, and device to deliver recommendations pertinent to the user’s present position. It can improve the user experience by taking into account the exact environment in which the recommendations are presented [22,23].**Knowledge-Based Filtering:** The KBF method suggests things based on clear user preferences and needs. It considers information supplied by the user, such as specific interests, desired qualities, or limitations, and recommends things that meet those requirements. It does not rely primarily on demographic information but instead on user-specified choices [24,25].**Demographic Filtering System:** DF determines user categories by employing demographic data such as gender, educational background, age, and so on. It does not have the new user issue because it does not use ratings to create suggestions. However, due to internet privacy concerns, it is difficult to obtain enough demographic information that is necessary today, limiting the use of DF. It is still used in conjunction with other recommenders as a quality-enforcing strategy [26].**Hybrid Recommender Systems:** HRSs integrate various recommendation methods to create a more accurate and personalized recommendation system. By combining more than one recommender system approach, hybrid recommender systems leverage multiple sources of data and algorithms to enhance the quality of recommendations. The goal is to reinforce the benefits of each strategy while minimizing their downsides or limitations, resulting in a more effective and comprehensive recommendation approach [13,15,16,18,21,27,28,29,30,31,32,33,34,35,36].

The goal of the study was to examine recently released HRS papers that focus on e-commerce and demonstrate the evolving perspectives of these systems, specifically their types, approaches, algorithms, and implementations in detail. Our key findings and contributions might be summarized as follows:-Data scarcity is a major limiting factor in the performance of recommendation systems. The current approaches to dealing with cold-start concerns for new users and objects frequently fail to incorporate demographic information into the suggestion process. According to research, the existence of cold-start users, together with the volume and quality of the surrounding data points used in the recommendation framework, have a substantial impact on prediction accuracy.-The contribution provides a synthesis of the existing information and approaches to hybrid-based quality in recommender systems via a thorough examination of the literature. This includes exploring, evaluating, and categorizing diverse hybrid models, assessment criteria, and real-world implementations, as well as identifying their strengths and drawbacks.-Identifying Challenges and Opportunities: The review recognized and articulated the distinct problems and opportunities provided by big data in recommender systems. This involved understanding the unique characteristics of big data, such as volume, velocity, and diversity, as well as the implications for hybrid recommender system design and implementation.-Proposing Frameworks and Rules: Drawing on the findings of the literature review, the contribution provided frameworks, architectures, or rules for designing and evaluating hybrid recommender systems in the context of big data. These frameworks incorporated best practices, addressed frequent hazards, and proposed solutions for dealing with big data’s distinct characteristics and requirements, such as scalability, real-time processing, and data integration.-Domain-Driven Insights: The review investigated the use of hybrid recommender systems in big data situations. It examined successful implementations in e-commerce, social media, healthcare, IoT, and the less-explored area of talent pool optimization for recruitment solutions.-Employing an open-source program, ASReview uses active learning to improve the systematic selection process in research. It efficiently processes vast amounts of text, reducing the number of documents that must be examined by humans and eliminating false negatives.

The following portions of the essay are organized as follows. Section 2 provides the related work and background. The objectives and reasons for conducting a systematic literature review are presented in Section 3. Section 4 describes the methodology for the review process, including the information sources, eligibility criteria, and data extraction, while Section 5 covers the synthesis of the results and discussion. Section 6 brings the paper to its conclusion.

The selected papers are presented at the end of this paper (see Appendix A).

## 2. Background and Related Work

The overabundance of irrelevant information frequently results in a significant investment of time and resources in the search for useful information, or possibly the inability to locate the necessary knowledge completely. Recommendation systems (RSs) have been created to address these difficulties. Their goal is to reduce these concerns by making specific recommendations and solutions.

The research in [37] emphasizes the value of alternate evaluation metrics for recommendation systems (RSs) in the classifieds area, in addition to typical accuracy measures. The key metrics that were discussed include the following:**Diversity:** Assesses the diversity among the recommendations, which is critical for providing users with a wide range of options and improving engagement. The paper in [37] uses measures such as test coverage, Shannon entropy, and the Gini index to assess diversity, with values 0.74, 10.40, and 0.79, respectively. Greater diversity in recommendations could offer consumers additional choices, potentially increasing user engagement and satisfaction.**Novelty:** Determines how surprising the recommendations are, which helps to keep users interested by suggesting goods they may not have considered.**User Satisfaction:** Assesses the total user experience using feedback and engagement metrics to customize suggestions to user preferences. By adding these indicators, HRSs can improve their performance, better correspond with user needs, and increase overall engagement and satisfaction.

The success of recommender systems is measured using a range of metrics that go beyond ordinary accuracy measures (accuracy, precision, recall, and F1-score). In practical implementations, it is critical to connect these measures with user-centric goals to ensure that the recommendations not only perform well algorithmically but also boost user happiness and engagement. In addition to the new metrics provided above (see Section 2), below is a thorough study of the alternative metrics used to evaluate recommender systems, especially in real-world applications. The papers in [38,39] covered different criteria for evaluating the efficacy of the Conformity-Aware Multi-Task (CAM2) model in the context of system recommendations and scoring hotels in the suggested recommendation system.

**Aggregated User Engagement:** This indicator measures how engaged users are with the system’s recommended content. The CAM2 model significantly increased this measure by 0.50%, demonstrating improved user involvement with the platform [39].**Daily Active Users (DAUs):** This indicator counts the number of unique users who interact with the site each day. The CAM2 model led to a 0.21% rise in DAUs, indicating that more users are returning to the site due to better suggestions [39].**Retention Metrics:** Renewal metrics are used to assess the model’s capacity to improve user experience and motivate return visits, particularly among casual users. The model’s design promotes better engagement and retention among casual users [39].**Reviews and Comments:** The system evaluates customer reviews to measure thoughts and sentiments about hotels, which assists in creating recommendations according to user preferences [38].**Surrounding Environments:** It considers surrounding Points of Interest (POIs) to assess the facilities accessible around the hotels, which can impact a user’s decision [38].**Numerical Ratings:** The system integrates numerical ratings submitted by users, serving as a quantifiable assessment of hotel quality.**Aggregated Scores:** The suggested system aggregates scores from both reviews and surrounding facilities, enabling a thorough evaluation of each hotel [38].**Polarity Ratings:** The system creates polarity ratings from reviews using natural language processing (NLP) techniques, which helps to comprehend the sentiments represented in the reviews [38].

The main goal of the research by Sivasankari et al. [40] is to create a hybrid scientific article recommendation system that uses the COOT optimization algorithm to improve the accuracy and relevance of article suggestions. The COOT optimization technique is intended to efficiently traverse the citation graph and discover highly important publications. The study addresses key issues in recommendation systems, such as the cold-start problem and user interest unpredictability, by combining content- and graph-based recommendation algorithms [40]. The COOT optimization algorithm is used to select articles that closely match user queries, ensuring that recommendations are highly personalized and matched to individual needs. The suggested strategy seeks to increase important performance indicators, such as precision, recall, and mean reciprocal rank (MRR), thus boosting the overall effectiveness of the recommendation system. Furthermore, the qualitative results show that providing more relevant and diverse recommendations increases user happiness, demonstrating the system’s effectiveness in satisfying users’ particular needs.

The article in [41] discusses how different amounts of novelty and variety in recommendation algorithms affect user happiness, algorithm performance, and system accuracy:**Impact of Diversification on User Satisfaction:** According to the study, user satisfaction is highest when recommendations have a balanced level of relevance and diversity, especially a diversity score of 0.6. This balance indicates that people respect moderately diversified content in their suggestions [41].**Relevance–Diversity Trade-Off:** One important point raised is the inevitable trade-off between relevance and diversity; as diversity grows, relevance frequently declines, potentially affecting user experience. This tension is critical for recommendation techniques that try to enhance both elements concurrently [41].**Algorithm Performance:** Algorithms that use a greedy, marginal relevance maximization (MMR) approach perform better in terms of diversity without compromising too much relevance. Adaptive algorithms that modify the timing of diversification outperformed similarity-based techniques [41].**Empirical Comparisons Using Metrics:** The article examines algorithms based on metrics such as ERR-IA and subtopic recall to assess relevance and variety. These measurements, especially when applied to movie genres, provide a complete picture of algorithm effectiveness [41].

The study on the kernel-mapping-based Group Recommender System (KGR) by Guo et al. [42] aims to enhance recommender system performance by addressing cold-start and data sparsity issues. The KGR model leverages user-trust relationships to form user groups, mitigating these problems. The study introduces kernel mapping techniques to create group kernels and matrices, enabling multilinear mapping between group–item interactions and user preferences. A hybrid model is proposed that combines group and individual user kernels, emphasizing individual preferences within groups. The KGR model [42] is validated on two trust-based datasets, demonstrating effectiveness through RMSE metrics. Optimal parameter values are identified to further improve the model’s performance and reduce RMSE errors. These strategies collectively enhance the accuracy and effectiveness of group recommendations in the KGR system.

The recommender system is a mechanism that helps users to make decisions in complex information contexts [3,14]. In the world of e-commerce, it is a tool that helps consumers to find knowledge that is relevant to their interests and preferences [29]. It also promotes the social process of relying on recommendations from others when personal knowledge or experience is insufficient. Recommender systems address the issue of information overload by making individualized and specialized recommendations for content and services. These systems have been designed using a variety of approaches, including collaborative filtering, content-based filtering, and hybrid filtering [13,30]. Collaborative filtering is the most widely utilized of these approaches. It recognizes people who share similar likes and recommends products based on their assessments.

Collaborative filtering has been applied in a variety of sectors, including news-based architectures, online social information filtering systems, and e-commerce platforms such as Amazon [43], Netflix [44], Spotify, YouTube, Facebook, news articles, and financial services [27]. On the other hand, content-based filtering associates content resources with user attributes, relying on human knowledge rather than the opinions of others.

Both collaborative- and content-based approaches provide numerous benefits, such as business advantages, personalization, efficiency, and discovery. However, they have some disadvantages, including limited content analysis, privacy concerns, a lack of user control, overspecialization, data scarcity, cold-start challenges, and scalability limitations. To address these restrictions, hybrid filtering methods have been proposed [15]. These approaches incorporate various filtering strategies to improve the accuracy and performance of recommender systems [29]. Hybrid filtering approaches are classified according to their operations: weighted hybrid, mixed hybrid, switching hybrid, feature-combination hybrid, cascade hybrid, feature-augmented hybrid, and meta-level hybrid [24]. Currently, collaborative filtering and content-based filtering methods are widely used, either by combining their predictions or adding features from one technique into the other [15,18,29,31,32,33,35].

In the study in [45], the authors investigated the several challenges of developing an effective hybrid recommendation system for online purchasing. The key issues are as follows:

Defining Lexical Variables: The fuzzy expert system uses linguistic variables to model ambiguous notions [45].

**Various Approaches:** The system includes collaborative filtering, content-based techniques, and a fuzzy expert system. It can be difficult to balance these many approaches and guarantee that they function together nicely, resulting in inconsistencies in ideas [45].

**Evaluating Performance Metrics:** Achieving great precision and recall is critical to the system’s performance. The study aspires for results above 90%, so comprehensive testing and validation against established methodologies is required to ensure dependability and effectiveness [45].

**User Choice Management:** The system must be able to react to changing user preferences and behaviors. This necessitates a reliable technique for capturing and evaluating user activity regarding online shopping, which can be challenging given the fast pace of customer interactions [45].

The primary goal of the study in [46] is to provide a hybrid recommender system designed to improve the selective dissemination of the research resources inside a Technology Transfer Office (TTO). The precise objectives described in the paper are the following:

**Improving Information Discovery:** The system is intended to assist TTO personnel and researchers in quickly locating relevant information, addressing the issues created by the expanding volume of available research materials [46].

**Personalized Suggestions:** The goal is to provide tailored suggestions based on user profiles, boosting the relevancy of the information supplied to users [46].

Facilitating Cooperation: The system is designed to detect possible cooperation opportunities among researchers, hence encouraging the formation of multidisciplinary teams to better research outputs.

**Using Fuzzy Lexical Modeling:** The article discusses the use of fuzzy linguistic modeling to describe qualitative information, which improves user–system interaction and the complete efficacy of the recommender system [46].

In the realm of recommendation systems, hybrid-based quality recommender systems are becoming increasingly significant. Combining several approaches has shown promise in raising the efficacy and accuracy of recommendations in a range of fields. The increasing need for customized recommendation services will surely require the research and development of hybrid-based recommender systems, which will help to reduce information overload and provide users insightful suggestions.

Hybrid recommendation systems based on quality consider both user preferences and the quality of the recommended goods at the same time, combining several techniques to deliver relevant recommendations [18]. With the goal of overcoming the drawbacks of single-strategy techniques, these quality-aware hybrid recommender systems offer an exciting evolution in the industry. These systems combine several techniques to provide more accurate and nuanced recommendations. These tactics include content-based filtering, which focuses on item attributes, and collaborative filtering, which leverages the behavior and preferences of other users. These hybrid systems’ capacity to manage the complexity and diversity of user preferences and item characteristics is one of their main advantages. For instance, a hybrid approach to movie selection may take into account both the user’s favorite genres and the films’ critical reception, guaranteeing that only well-received films are recommended.

## 3. Goal of the Literature Review

Systematic reviews use rigorous and transparent procedures to provide a full and impartial appraisal of several relevant studies in a single document. A systematic review’s goal is to synthesize and summarize the current body of knowledge, with the goal of uncovering all the relevant data relative to a certain subject. It is an additional area of study that aims to locate, evaluate, and interpret all the available information from primary studies that is relevant to a specific research issue. To guarantee a robust and systematic literature review (SLR) approach, we followed the standards stated in the Cochrane Handbook [47] and those proposed by Kitchenham and Charters [48,49]. These criteria, which are widely accepted in the research community, provide a foundation for conducting comprehensive and unbiased assessments. We aimed to reduce bias and ensure the reliability and validity of our systematic review by adhering to these established principles (see Figure 1).

The overall goal is to assess the progress of hybrid recommender techniques and propose potential topics for further study. The objectives are to examine the current trends in difficulties, approaches, datasets, application areas, and assessment measures using a hybrid approach. A systematic literature review is a time-consuming task that requires the researcher to design the protocol, adjust the search string, filter the results, sometimes more than a thousand articles, select those that meet the inclusion criteria, and remove those that do not meet the exclusion criteria. Following that, the researcher may begin to study the relevant results one by one.

### 3.1. Reasons for Conducting Systematic Literature Reviews

A systematic literature review is performed for a variety of reasons [48]:Summarizing the existing knowledge and information concerning research questions or technology, such as the empirical evidence on the benefits and limitations of a specific agile approach. They provide a comprehensive overview of what is known in the field.Identifying Knowledge Gaps: Systematic reviews can discover knowledge gaps by reviewing existing material. These gaps can assist researchers in identifying places where further study is required.Making Choices Based on Proof: Systematic reviews are an important tool for making evidence-based decisions. They serve as a foundation for making educated judgments in a variety of disciplines, including healthcare, education, and policy creation.Minimizing Bias: Systematic reviews locate and choose relevant research in a systematic and accessible manner. This decreases the possibility of bias in study selection and interpretation, making the results more credible.Bringing Conflicting Evidence Together: In some domains, the literature may present contradictory conclusions. Systematic reviews seek to synthesize and evaluate contradictory material to present a more complete picture of the state of knowledge.Policy and Practice Insights: Systematic reviews are frequently used to inform policy decisions and clinical practice guidelines. They provide a solid evidence framework for making recommendations and judgments with substantial societal implications.Time and Resource Efficiency: Conducting a systematic review might be more efficient than beginning a new study, especially if the issue has previously been well investigated. By using the current research, it can save time and resources.Systematic reviews can aid in the prevention of duplication of research efforts. Researchers may assess what has previously been completed and concentrate their attention on areas that require fresh investigation.Establishing a Baseline: A systematic review can serve as a starting point for researchers who are new to a topic, offering a baseline grasp of the present state of knowledge. Systematic literature reviews, on the other hand, can be used to assess how much the empirical data supports or contradicts the theoretical assumptions, or even to aid in the development of new theories.

### 3.2. The Value of Systematic Literature Reviews

The goal of systematic reviews is to synthesize available knowledge in an equitable and transparent manner. They adhere to a preset search technique that allows them to analyze the completeness of the search. Researchers performing a systematic review must seek out and publish findings that both support and contradict their favored study hypothesis. Systematic reviews improve the integrity and credibility of the research process by adhering to these standards.

They are critical tools for advancing knowledge, influencing decision-making, and guaranteeing the use of the best available evidence in a variety of research and practice sectors.

## 4. Methodology for Review Process

The Preferred Reporting Items for Systematic Reviews and Meta-Analyses (PRISMA) guidelines comprised the methodology used for this investigation [50]. Several recommendation techniques have been proposed and applied in the field of recommendation systems. However, implementing these techniques in the context of hybrid recommendation systems poses several challenges and opportunities while considering quality. The primary objective of this research is to examine the current and emerging approaches applied to hybrid recommendation systems in the recent research literature and outline avenues for future research. To ensure a systematic review process as indicated above, we have adopted the guidelines from [47,48]. The steps of our review process are illustrated in Figure 1. It involves eight main steps: research question formulation, establishment of systematic review protocol, performing an extensive literature search, screening and selecting studies, examining the bias and quality of the studies, data extraction, analyzing the information gathered, and sharing the outcomes (see Figure 1).

### 4.1. Question Formalization

The fundamental purpose of this systematic literature review is to learn what difficulties HRSs could successfully handle, how they are built and evaluated, and how they could be experimented with in terms of manner or features [24,51,52]. Thus, the following research questions (see Table 1) were developed:

To address these research questions, we generated a research string utilizing terms related to our topic.

The primary keywords are hybrid, quality, recommender systems, dissemination, information, and big data, and then we introduced synonyms to obtain the final list of keywords, as shown in Table 2.

We employed Boolean operators in our systematic literature review search method. These operators, which include “AND” and “OR”, are used to connect alternative terms. We can cluster synonymous or related phrases by using the “OR” operator, and we can merge distinct components inside the search string by using the “AND” operator. We created a comprehensive and precise search string by skillfully applying these operators, allowing us to highlight relevant studies and gather valuable insights for our methodical assessment of the literature. Then, we used the selection strategy, which was based on some critical factors such as the year of publication, the language of the paper, and the title. We restricted our research to English papers. In addition, we considered the reputation and validity of the journals, as well as the recently published papers. Subsequently, we reviewed each item and selected those that were relevant to the topic. As a result, the selection procedure consists of three major steps: searching, paper and journal filtering, and content-based selection.

Consequently, we obtained the study string illustrated in Table 3.

### 4.2. Database Analysis and Research Methodology

Researchers have studied hybrid recommender systems in many different studies. We examined this research using the established method of a systematic literature review, which is based on the state-of-the-art recommendations outlined above.

This protocol’s steps are as follows (see Figure 2):Identifying research questions.Previous research findings.Searching databases for relevant research based on hypotheses.Selecting data based on predefined inclusion and exclusion criteria.Analyzing the collected data.Study findings.Ideas for further research.

The information sources chosen are most prominent scientific databases that have been used for other relevant works of indexed journals in Journal Citation Report (JCR) [3,7]. These datasets are as follows:**Scopus**: This database may accept the complete query and offers the options to specify additional particular filters (see Table 4 for question formulation and Figure 3 for yearly distribution).**Web of Science:** When conducting a systematic review, employing Web of Science has benefits like thorough coverage, easy access to high-quality content, citation tracking, sophisticated search tools, effective bibliographic management, and collaboration support. Using these elements, systematic reviews can be made more thorough and rigorous, enabling researchers to efficiently find, assess, and synthesize pertinent papers (see Figure 4).**Springer Link:** To comply with the download limit of 1000 objects imposed by this database for the csv file, filters must be added to the item list. Because the current query returned 4068 items at that time, exceeding the allowed threshold, it was critical to narrow the list. This can be accomplished by implementing filters that consider criteria such as publication date, discipline, language, and content type. By incorporating these filters, we could effectively reduce the item list while still adhering to the download restriction (see Figure 5 for distribution by year).**Google Scholar:** To overcome the limitation of extracting Google Scholar search results, we used the open-source tool “Harzing’s Publish or Perish” for exporting the results in Excel (see Table 5). The search process in this database presented additional challenges when compared to other databases for three main reasons:
Incomplete Search String: Google Scholar does not allow you to directly enter a complete search string. As a result, we had to use the basic search tool to conduct a search that would return results matching the initial search string.Difficulty in Search: Google Scholar’s search functionality is more intricate than that of other databases, making it more difficult to obtain desired results. To retrieve the relevant information, careful navigation and the use of appropriate search techniques were required (see Figure 6). In fact, due to the absence of results from the original query, we decided to cancel it in order to address this issue. In its place, we shall investigate an alternative technique by running the below query.However, using this specific database made it possible to include “gray literature”, including proceedings from conferences.**ACM Digital Library** https://dl.acm.org/, access date: 10 December 2023 Because of the enormous quantity of papers, we narrowed our search to the years 2020 to 2024 and focused on journals (see Figure 7).

### 4.3. Eligibility Criteria

As previously stated, the fundamental goal of a systematic review is to collect relevant techniques suggested within a certain field. To guarantee that only relevant articles are kept during the search process, the inclusion and exclusion criteria for a literature review must be properly defined [48,53].

The specific traits, attributes, or criteria that are utilized to determine whether a given study or article should be included in the review are referred to as inclusion criteria (IC).

Exclusion criteria (EC), on the other hand, are the specific features, attributes, or conditions used to determine which studies or papers should be rejected during the review process. A paper is considered to be eligible if it meets the following requirements:**IC1:** Papers offering hybrid quality-based recommender systems, algorithms, and techniques in the context of big data.**IC2:** Papers from conferences and journals published between 2020 and 2024.**IC3:** The paper incorporates search-relevant keywords within its title or abstract.**IC4:** The paper addresses hybrid recommendation systems.**IC5:** The paper addresses at least one problem of recommendation or proposes at least one technique of hybridization.

The exclusion criteria are the following:**EC1:** The publication date is earlier than 2020.**EC2:** The paper is written in a language other than English.**EC3:** The paper is a short article, a standard, a poster, an editorial, or a tutorial.**EC4:** The title, abstract, and keywords are not relevant to the research topic.**EC5:** The paper does not discuss hybrid recommendation systems.

### 4.4. Information Sources

In accordance with review process Step 2 (see Figure 2), we ran the search string through the search engines of some digital libraries, yielding a total of 5857 preliminary primary studies (see Table 6). This retrieval process was conducted at the start of 2024. The varying number of publications obtained from digital libraries is due to changing the primary query (see Figure 8) of the search in certain databases that have a limit on the number of Boolean operators, using third-party data extraction tools, and differences in search engine filtering settings. We developed a set of inclusion/exclusion criteria, as shown in Section 4.3, to help us make rational decisions about which exploratory investigation to pursue further. These requirements serve as the foundation for focusing on the most relevant research that aligns with the review’s aims. Duplicate papers were removed, and a coarse selection phase followed. Given the impracticality of processing all publications, we decided to include just journal articles, scientific articles, and machine learning articles in some databases and all article types in others, excluding workshop presentations, review reports, and gray literature, especially for Scopus and Springer data, due to the number returned by the basic query (see Figure 8). We started by looking at the title, publishing type (conference, workshop, journal, etc.), and publication year. We looked at the abstract or other sections of each article in many situations to determine its relevance. Because the goal of this review study is to focus on quality in a large data setting with hybrid recommender systems, we chose articles that offered mixed or blended recommendation systems while avoiding those that addressed single recommendation techniques or did not discuss recommendation systems at all in the context of big data.

The first selection process, along with the application of date-related inclusion and exclusion criteria, yielded a list of 3557 articles. Following that, we conducted a more thorough review and selection of articles, limiting ourselves to specific sorts of articles to yield 131 articles. Following that, we conducted a more in-depth review and selection of the papers, selecting only open-access articles, to yield 81 articles. (see Figure 8). The whole list, as well as publication information, can be found in Appendix A.

As indicated in Figure 8, these statistics provide an overview of the number of articles discovered in multiple databases and based on various search parameters, such as publication date, article type (e.g., journal articles, open access, etc.), and specific topic (e.g., science, machine learning, and English). These data will be used to refine our search or analyze the relevancy of the results based on our individual research objectives.

Here is an interpretation of the numbers indicating the articles found in various databases and using various search criteria:**Preliminary Research Findings** (see Figure 8):Total articles found in the preliminary research: 5857 articles.**ACM:**Total articles found in the ACM database: 376 articles.Articles from 2020 or later in the ACM database: 187 articles.Total journal articles found: 33 articles.Total open-access articles found: 19 articles.**Google Scholar:**Total articles found on Google Scholar: 55 articles.Articles from 2020 or later on Google Scholar: 13 articles.Total articles of all types on Google Scholar: 13 articles.Open-access articles on Google Scholar: 6 articles.**Scopus:**Total articles found in Scopus with basic query string: 1348 articles.Articles from 2020 or later in Scopus: 838 articles.Total articles with restrictions (English, engineering, ML, business, etc.): 28 articles.Open-access articles in Scopus: 14 articles.**Springer:**Total articles found in Springer: 4068 articles.Articles from 2020 or later in Springer: 2509 articles.Articles related to science in Springer: 32 articles.Open-access articles in Springer: 32 articles.**Web of Science:**Total articles found on Web of Science: 10 articles.Articles from various dates in Web of Science: 10 articles.Total articles of all types in Web of Science: 10 articles.

### 4.5. Data Extraction

At this point, every study that was part of the systematic review had been located, and we need to move on to extracting the data. A template can be used to gather the data needed to analyze the studies. Standard documents are available for this purpose, such as the Preferred Reporting Items for Systematic Reviews and Meta-Analyses (PRISMA) Statement [54] and the Cochrane data collection form for intervention review [55]. These forms can be utilized in the training and education sciences, and analysts can modify and test them in accordance with the goals of the systematic review.

During this phase, we designed a customized form with a range of parameters such as title, author, year, and so on [48]. The form was then filled out with information about the research topics for all the selected papers. Table 7 contains a list of these attributes. The purpose of this operation was to collect and synthesize data to answer the defined research questions. The extracted data were listed in the first column, an explanation for some of the extracted data that may appear ambiguous is provided in the second column, and the research question to which the data are connected is provided in the third column (see Table 7).

## 5. PRISMA Checklist

The PRISMA Checklist is a tool used mostly in the field of health and research to assess the quality of studies and reports (see Table 8). The term “PRISMA” refers to the abbreviation “Preferred Reporting Items for Systematic Reviews and Meta-Analyses”.

### 5.1. Main Objectives

**Transparency:** Ensure that systematic studies and meta-analyses are presented clearly and completely.**Quality:** Improve the quality of research reports to facilitate understanding and evaluation.**Standardization:** Provide a standardized framework for researchers to follow while writing their work.

### 5.2. Components

The PRISMA Checklist often includes a list of important criteria to follow, such as the following:The definition of research objectives.The methodology for selecting studies.Evaluation of bias.The synthesis of results.

### 5.3. Utilization

Researchers use this checklist to ensure that they cover all the necessary aspects while writing their studies, which contributes to better research valorization and use in the scientific environment.

## 6. Results Synthesis and Discussion

In this section, we provide the findings from the selected studies, addressing the research questions (see Section 4.1) by analyzing categorized challenges, procedures, hybridization classes, and evaluation methodologies. The investigations highlight a variety of concerns, including information overload, suggestion accuracy, and system scalability, all of which offer substantial hurdles for e-commerce recommendation systems. These identified concerns inform our investigation of potential approaches for overcoming them. To address these issues, the research applies to a variety of recommendation strategies, including collaborative filtering, content-based algorithms, and deep learning approaches. Each strategy is designed to address a certain need, such as increasing customization or lowering computational demands. This diversity emphasizes the necessity of using the proper strategy for each individual challenge, and examples from the research show their success in various aspects of e-commerce.

We assessed the numerous papers that we deemed relevant for our evaluation from various angles.

We used examples from the included research to show the various kinds of problems, strategies, hybridization classes, assessment methodologies, and so on.

### 6.1. Quantitative Evaluation

This part examined the screened papers in the hybrid recommender systems, concentrating solely on three types of metadata: database, year of publication, and information sources. To address these issues, the research applies a variety of recommendation strategies, including collaborative filtering, content-based algorithms, and deep learning approaches. Each strategy is designed to address a certain need, such as increasing customization or lowering computational demands. This diversity emphasizes the necessity of using the proper strategy for each individual challenge, and examples from the research show their success in various e-commerce contexts.

#### 6.1.1. Data Origin

The percentage of papers in each database is shown in Figure 9a. We found nineteen papers in the ACM database (24%), six papers in Google Scholar (7%), fourteen papers in Scopus (17%), thirty-two papers in Spinger (40%), and ten papers in Web of Science (12%). With a percentage of 40%, we observe that Springer offers the repository with the highest quantity of papers. Springer’s extensive collection may be due in large part to the company’s lengthy history, well-established reputation, and fame in academic publishing.

#### 6.1.2. Year of Publishing

As previously mentioned, the research was carried out for the period 2020 to the beginning 2024 excluding Web of Science and Scopus, which include data from all dates (until the beginning of 2024). The diagram presented in Figure 9b shows the number of papers by year of publication.

Additionally, the pie chart in Figure 9b reveals that the current year has the fewest articles. This outcome is understandable given that the data were taken at the start of 2024. The timing undoubtedly adds to the lower figure as it does not account for possible publications that may appear later in the year.

Below is a PRISMA flowchart that illustrates the inclusion and exclusion strategies used in the study (see Figure 10). This flow chart was chosen for this study, describing the flow of information through the various phases of a systematic review. It shows the number of records identified, included and excluded, as well as the reasons for exclusions. This section focuses on the four inclusion and exclusion stages of the PRISMA table: identification, selection, eligibility, and inclusion. The search engine results from the five databases (ACM, Google Scholar, Scopus, Springer, and Web of Science) yielded a total of 5857 articles. More than 2300 publications were eliminated because they did not correspond to the time range set over the past 4 years for databases that return many articles and over the past 10 years for those that return few. For certain databases, such as Scopus, which are very consistent, we limited the search to journal, computer science, machine learning, and business and management articles in order to reduce the scope of this study. We obtained one-hundred-thirty-one documents. We then eliminated fifty publications due to a lack of full text, four articles due to redundancy, and twenty-five others because they did not correspond to the subject’s relevance based on their titles, keywords, or abstracts, or because they did not meet the eligibility criteria, either because their text was not directly related to our field of research or because their content lacked detail and precision, resulting in fifty-two documents at the end.

The interaction of technology improvements and publishing patterns in hybrid recommender systems demonstrates how fast-changing tools and approaches can influence research paths. As neural network architectures evolve, transformer models gain popularity, and privacy concerns grow, researchers respond with novel answers and techniques. These variables not only explain the increase in publication numbers but also indicate a dynamic area that is evolving to meet the demands of the current technology and societal needs. The number of articles on recommender systems over time shows a significant trend (see Figure 11). This graph depicts the values after making various PRISMA analysis selections, particularly the inclusion and exclusion criteria. Several significant patterns emerge from the publication trends over time, reflecting the evolution of study interest in hybrid recommender systems and associated technologies. Starting with modest research production in the early years, such as 2004, 2008, 2012, and 2018, with only one publication in each of these years, it is obvious that the topic was in its early stages or garnered little attention during this time. Evidently, the field was still in its early stages of development, which could be explained by a lack of funding and interest. However, 2020 was a watershed moment, with the number of papers jumping to seven, indicating increased interest or developments in the field. This rising trend continued into 2021, when the count increased to eight items, then surged again in 2022, reaching fifteen.

In 2023, the total rose slightly to sixteen, indicating a continued rise in research production. This consistent rise in recent years indicates a growing recognition of the value and relevance of recommender systems in academic discourse. These rapid increases can be attributed to the emergence of deep learning techniques, which enabled the integration of complex data representations into recommendation models, as well as the frequent adoption of cloud computing and technologies for big data, which made it easier to manage large-scale, multifaceted data. The focus of the academic community on enhancing user experience, combined with the industry’s quest for more personalized and adaptive recommendation engines, are likely to have fueled this development. Using transformer models could potentially help to hasten this advancement. The introduction of transformer models was a significant milestone in natural language processing (NLP) as well. The findings show that scholars are increasingly interested in this area owing to its practical applications and theoretical significance. As the research in this area progresses, it is critical to monitor these trends in order to understand the changing environment of recommender systems. Overall, this tendency implies that a vibrant and dynamic field is gaining traction within the academic community.

Figure 11 and the pie chart in Figure 9b show that the current year has the fewest number of articles. This development is expected given that the data were collected at the beginning of 2024. The time of data collection most certainly influenced the outcome, resulting in fewer publications being available this year. The early collection period does not accurately reflect the possibilities for publication throughout the year. This understanding is critical for appropriately analyzing the data. The figures emphasize the seasonality of article creation. As a result, the current figures should be considered in light of their chronology. Overall, the findings point to a transient dip rather than a long-term deterioration. Future analysis may provide a more complete picture when additional articles are released. Given the observed patterns in the publication numbers, it is critical to investigate how key technological advancements influenced the landscape of hybrid recommender system research. Advancements in neural network designs, the adoption of transformer models, and the increased emphasis on privacy-preserving strategies have all likely had a significant impact on the publication trends. We will examine each of these elements in depth:
**Advances in Neural Network Architectures:** Recent years have witnessed tremendous advances in neural network topologies, which have transformed the field of machine Learning and, by extension, recommender systems.
**Deep Learning Techniques:** The development and refinement of deep learning approaches have enabled academics to develop more sophisticated models capable of processing complex data inputs. These developments enable better representation learning, in which models can automatically recognize patterns and features in raw data, resulting in higher recommendation accuracy.**Hybrid Approaches:** The merging of several neural network architectures, such as convolutional neural networks (CNNs) for image data and recurrent neural networks (RNNs) for sequential data, has aided in the creation of hybrid recommender systems that can use numerous data sources. This flexibility is most certainly a major contributor to the current increase in publication rates.**Adoption of Transformer Models:** Transformer models have ushered in a new era of natural language processing (NLP) and beyond.
**Transformer Architecture:** Transformers, introduced through models such as BERT and GPT, have raised the bar for comprehending and creating human language. Their capacity to capture long-term dependencies in data makes them ideal for jobs involving user interactions and preferences in recommendation systems.**Impact on Recommendations:** The potential to more effectively simulate user behavior and preferences with transformers has prompted study into their use in recommender systems. This has most likely led to the rise in publications as academics investigate creative ways to integrate transformers into hybrid models, increasing their effectiveness across many domains.

To obtain a second opinion on this study, we used Active Learning for Systematic Reviews (ASReview) as a secondary reviewer to identify the relevant articles [56]. This tool is a machine learning software that implements different machine learning algorithms that interactively query the researcher (see Figure 12). It enables the systematic review of articles and analysis of metadata. ASReview could significantly improve the efficiency and relevance of the systematic literature review process. ASReview allows the user to sort documents while the active learning algorithm (Naïve Bayes by default) ranks unlabeled documents in the background, from most relevant to least relevant.

It is sometimes viewed as a tool for selecting titles and abstracts in systematic reviews or meta-analyses, but it can handle any type of textual data that needs to be selected systematically.

Using the AI tool “ASReview” required multiple steps [57]. Before screening, the software required training for its algorithm with multiple prelabelled papers. The AI tool then offered the article with the greatest chance to be relevant using a researcher-in-the-loop approach. The reviewer then determined the relevance of each recommended article. This procedure was repeated until the stopping requirement was met.

The objective is to screen less data than are in our dataset, and simulated research has shown that we may skip up to 95% of documents [56], although this is extremely dependent on the dataset and inclusion/exclusion criteria [58]. When we have decided to finish screening, we may export the findings (i.e., the partially labeled data and the project file with the technical information to replicate the entire process) and post them on sites like the Open Science Framework. Finally, in ASReview, mark the project as completed.

ASReview LAB saves time, improves the quality of results, and makes work more transparent when examining large quantities of textual data to extract the relevant information. Active learning will facilitate decision-making in any discipline or industry.

Using the AI tool involved multiple stages, prior to screening, the tool’s algorithm needed to be trained using several prelabelled articles [57]. Next, using a researcher-in-the-loop approach, the AI tool recommended the article with the highest likelihood of relevance. The reviewer then determined the relevance of each article proposed. The operation was repeated until the halting requirement was met. All papers deemed relevant by the reviewer were reviewed for full text (see Figure 12).

The following are the essential steps [59]:**Data Import:** Import the entire set of research documents into the ASReview software (that is, the metadata containing the text of the titles and abstracts).**Initial Formation:** ASReview begins with an initial formation phase. The researcher classifies a small subset of articles as relevant or irrelevant in order to form the automatic learning model. In fact, prior knowledge is chosen and used to create the first model and present the first recording to the researcher. Because this is a binary classification problem, the evaluator must choose at least one key record to include (specify label: relevant) and at least one key record to exclude (specify label: irrelevant) based on prior knowledge. An automatic learning classifier is tasked with predicting the relevance of the study (labels) based on a representation of the text containing the recording (characteristic space) and prior knowledge.After being trained with previous expertise, the AI tool ranks all unlabeled papers (i.e., articles that had not yet been determined to be eligible) from highest to lowest probability of relevance [57].To avoid any authority bias in the inclusions, we have purposefully chosen not to include the name of an author or a representation of a network of citations in the space for characteristics.**Active Learning:** ASReview employs an active learning strategy. The model examines the labeled articles and selects the most ambiguous or informative ones. These articles are presented to us in order to manually examine and categorize. Alternatively, during the active learning cycle, the software displays a new record that the user must examine and label. The user’s binary etiquette (1 for relevant and 0 for irrelevant) is then used to create a new model, after which a new record is presented to the user. This cycle will continue until the user specifies an end point.Currently, the user has access to a file that contains (1) entries that have been labeled as relevant or irrelevant and (2) entries that have not been labeled but are likely to be relevant based on the current model’s predictions [56].This configuration allows us to search for a large dataset much faster than possible with a manual process while maintaining decision-making transparency.**Iterative Process:** the researcher examines the selected articles and assigns labels (relevant or not). ASReview incorporates the labeled data into the overall training and updates the automatic learning model.**Model Refinement:** The updated model learns from our labeled data and improves its ability to predict the relevance of unlabeled items.**Iteration:** Steps 3–5 are iteratively repeated. The model continues to select new articles to investigate based on its uncertainty, and the researcher labels them in order to refine the model. This iterative process reduces the number of articles to be manually examined while maintaining high precision.**Final Article Selection:** When the model reaches a stopping point (for example, a desired level of examination exhaustion), ASReview returns a list of articles classified according to their predicted relevance. This list will assist us in focusing our attention on the articles that are most relevant to our systematic review.

Using ASReview, the researcher can significantly speed up the selection process by assigning priority to the most relevant articles for the examination while reducing the number of irrelevant articles that must be evaluated manually.

### 6.2. Out of Scope

The search mechanisms used in online databases are not perfect, so a substantial number of papers obtained during the first phase of the appraisal are unrelated to the searching scope. For this reason, a qualitative analysis founded on the examination and assessment of content is required (see Figure 10).

### 6.3. Qualitative Analysis

The selected articles were classified using fundamental recommender system approaches. Table 9 shows how we classified the relevant studies into different groups. Regarding relevancy according to inclusion/exclusion criteria, each study’s quality and completeness were considered (in terms of problem characterization, description of suggested method/technique/algorithm, and evaluation of findings).

The research we examined shows a substantial tendency toward the growth of hybrid recommender systems (HRSs). According to publication year, over 75 percent of the studies we reviewed were published within the last three years (see Figure 11). These statistics definitely suggest an increase in interest and research conducted in the field of HRS. Researchers and practitioners are noticing the potential benefits and advantages of integrating multiple filtering algorithms to improve the effectiveness and performance of recommendation engines. The expanding corpus of recent literature indicates that hybrid recommendation systems are becoming increasingly important and relevant in addressing the constraints and limits of classic single-approach recommendation approaches. This trend emphasizes the field’s dynamic character and ongoing efforts to develop more accurate tailored recommendation systems via the combination of various methodologies.

#### 6.3.1. Evaluation of Quality

A systematic review locates, evaluates, and critically assesses pertinent studies by applying explicit and systematic methods to a well-defined research question. Additionally, it gathers and arranges data from the studies to comprise the review. The results of the included studies are not always analyzed and summarized using statistical techniques (meta-analysis) [97]. The relationship between the research question, methods, results, and interpretation is assessed using a technique for evaluating the original quality of research using methodological quality protocols, checklists, and/or scales. As such, the validity and applicability of synthetic research findings depend heavily on the methodological quality of the original studies [97].

To estimate the quality of the chosen studies, we also developed the nine questions that are listed in Table 10.

We use weights of 0.5 for low importance, 1 for medium significance, and 1.5 for high significance to assign weights to the questions. These coefficients are essential in establishing how important each question is in relation to the others during the evaluation procedure.

Moreover, rate values are used to evaluate the answers to the questions. A “no” answer receives a score of 0, a “partly” answer receives a score of 0.5, and a “yes” answer receives a score of 1. These score values are useful for quantifying responses and evaluating study quality [3,24].

The following formula is used to explain each paper’s evaluation [3,24]:(1)$$Evaluation_{paper}=\frac{\sum_{i=1}^{N}q_{wi}*a_{ri}}{N}$$
which performs a product operation between the query weight ($q_{wi}$) (0.5, 1, 1.5) and the answer rating value ($a_{ri}$) (0, 0.5, 1). N = 9 in our case is the number of quality questions (see Table 10). Papers must meet the quality threshold of 0.80 in order to be accepted.

#### 6.3.2. Word Cloud and Frequency

Before creating the word cloud, stemming was used to discover the phrases’ common origin.

To begin the classification process, the tag cloud presentation was utilized to determine the major keywords. The clouds of the 30 keywords from the abstracts are provided in Figure 13 and Figure 14, which provide the 1000 important words from the whole texts of the articles with their relative relevance and prominence.

Keywords were analyzed using Python, NumPy, Pandas, and Matplotlib to produce a simple frequency analysis and word cloud graph.

Before constructing the graph, all characters in the text were converted to lowercase. Pre-processing included deleting digits, punctuation, and stop words often found in English.

Figure 15 represents the frequencies of the first 30 words extracted from the abstracts of all publications, while Figure 16 presents the frequencies of relevant words constructed from 1000 words selected from the content of all sections of the paper, omitting references.

### 6.4. Approach to Inclusion and Exclusion Standards

To make sure the studies chosen for analysis were pertinent, we used precise inclusion and exclusion criteria. We determined significance during the filtering process in the following ways:**Initial Retrieval:** After a retrieval process, 5857 preliminary primary studies were found using five digital libraries’ search engines. Each library utilized various filtering parameters, which resulted in differing quantities of papers being returned. Each library utilized various filtering parameters, which resulted in differing quantities of papers being returned.Criteria Definition: In order to concentrate on the most pertinent studies, we established a set of inclusion/exclusion criteria. Except for gray literature, workshop presentations, and articles that reported just abstracts or presentation slides, this involved choosing only journals for the Scopus and Springer databases and all categories for ACM, Google Scholar, and Web of Sciences. The chosen papers were to highlight current developments in the discipline and be published between 2020 and 2024.**Selection Based on Peer Review:** To ensure a degree of quality and credibility in the chosen studies, we only included articles that were approved for publication after a peer review procedure. Articles that were not peer-reviewed or did not fit the designated research focus were disqualified. Additionally, articles that did not include recommender hybrid techniques in their abstract or title were not included. This was essential for maintaining attention on the pertinent subject. To ensure linguistic and understanding consistency, non-English papers were eliminated.**Coarse Selection Phase:** We first examined the publishing type, year of publication, and title as part of our coarse selection phase. We frequently looked at abstracts or other sections of the publications to determine their applicability.**Hybrid Recommendation Systems:** The review excluded papers that had nothing to do with recommender systems and instead focused on those that presented hybrid recommender systems.**Data Entry and Analysis:** To enable a methodical review process, the data were input into an Excel spreadsheet, including keywords and cited information.**Quality Assurance:** To guarantee high-quality results, a systematic review uses a weighted score system to quantify study quality, accepting only those that meet a threshold of 0.80.**Final Selection:** Fifty-two primary papers that satisfied the predetermined standards were ultimately chosen, offering a strong basis for the systematic review. By guaranteeing that only pertinent and excellent papers were incorporated into the analysis, this exacting process raises the review’s academic worth and transparency.

### 6.5. Commonly Used Principal Summary Measures

In a systematic study of hybrid recommender systems, performance metrics are often employed as primary summary measures rather than standard measures such as risk ratios or odds ratios. In this scenario, summary measures would be used to assess the success of hybrid recommender systems. Some common performance indicators include the following:

**Precision:** The percentage of recommended items that are relevant.

**Recall:** The percentage of relevant items that are recommended.

**F1-Score:** The harmonic mean of precision and recall, which achieves a balance between the two.

These three metrics presume that the provided data are divided into “relevant” and “irrelevant” categories and may be organized into confusion tables (see Figure 17). The precision of a system is calculated by dividing the number of genuine positives by the total number of positive cases predicted by the system. The precision measure can be defined as the system’s precision in percentage terms using the generic confusion table. The recall value determines how well the system captures relevant instances and is calculated using the recall equation. The F1-score assesses the system’s accuracy and is calculated as the weighted average of the precision and recall scores. The findings for the hybrid recommender system are as follows:

Precision (0.80) indicates that 80% of the things recommended by the system are relevant. A precision of 0.80 is high.

Recall (0.92): A recall of 0.92 indicates that the system can retrieve 92% of the relevant elements, implying that it is quite effective at avoiding forgetting crucial recommendations. This high recall indicates that the system effectively covers a wide range of relevant articles.

F1-Score (0.86): The F1-score, which measures precision and recall, is 0.86. This rating shows great overall performance, implying that the system’s recommendations are generally accurate (high precision) and comprehensive (high recall).

### 6.6. Challenges and Setbacks (RQ1)

In response to RQ1, this section explains the different obstacles currently in use that recommendation systems face and offers different answers to these challenges.

#### 6.6.1. Approaches for Addressing the Cold-Start Problem

Cold-start was the most critical issue discovered. It becomes challenging when the recommender system is unable to draw any inferences from the little available data. Cold-start is a circumstance in which the system is unable to create effective recommendations for cold (or new) consumers who have rated no or only a few items. It typically happens when a new user enters the system or when new items (or products) are added to the database. Approaches to the cold-start problem usually concentrate more on gathering extra information such as user registration details or item metadata.

For this issue, the CF-based recommendation with an implicit rating was used in the study in [30]. Because explicit rating information on items was not available for online shopping malls, this method was used. The researchers extracted implicit rating information from transaction data, which served as a proxy for explicit rating information.

The authors of [29] created a hybrid recommendation system for personalized course recommendations in e-learning settings, which addresses cold-start difficulties and insufficient information.

Modern hybrid systems effectively incorporate several technologies, such as machine learning and deep learning, to address the user cold-start problem, outperforming previous systems that often rely on a single strategy. This integration enhances performance by combining data-driven and method-driven strategies [98].

**Meta-Learning:** Modern systems use meta-learning to quickly adapt to new users with less data, but traditional systems struggle to make recommendations without significant previous knowledge [98].

**Deep Learning Capabilities:** Hybrid systems commonly use deep learning techniques to capture complex interactions between people and things, which is a difficult task for traditional systems. This enables more tailored recommendations, even when less user data are available [98].

**Multiple-Feature Fusion:** Modern systems can combine a variety of features and data sources, enhancing their recommendation capabilities for new users. Traditional systems lack this adaptability and rely on simpler models that may not accurately reflect various user preferences [98].

To solve the cold-start problem in content-based recommender systems, effective user profiling is required. This can be accomplished by leveraging demographic variables such as geographic location, age, gender, occupation, and education [7]. One effective technique is to utilize onboarding questionnaires to collect user preferences at the beginning of program use [99]. This procedure entails connecting the initial data acquired to subsequent recommendations, thereby incorporating user preferences into the recommendation architecture.

Businesses that aggressively seek explicit feedback from new users via onboarding questionnaires, surveys, or interactive chatbots can acquire significant insights into client tastes and preferences from the start.

This direct approach not only improves the relevancy of recommendations but also contributes to the establishment of a personalized experience, thereby alleviating the issues connected with the cold-start issue [99].

#### 6.6.2. Sparsity

Approaches to the data sparsity problem concentrate more on using existing data to fill in the gaps. To make accurate recommendations, collaborative filtering (CF) requires many users who have rated many items. However, this is not always the case, resulting in sparsity issues. To address this issue, the paper [30] suggests a hybrid approach that combines CF with sequential pattern analysis (SPA). The limitations of CF in reflecting changes in user preferences over time can be reduced by integrating SPA, which considers item associations, with CF, which uses rating information. By providing recommendations based on both rating information and sequential patterns, this hybrid approach helps to mitigate the sparsity problem.

The combination of sequential pattern analysis (SPA) and collaborative filtering (CF) was used in [30] to address the sparsity problem. The study aimed to mitigate the higher probability of inaccurate and biased recommendations for items that arise from considering only purchasing information rather than rating information by integrating CF, which uses evaluating information, with SPA, which returns adjustments to user choices over time in a sequence of sequential patterns. The techniques of modern hybrid recommender systems and conventional systems are compiled based on the general recommendation framework. Reducing the dimensionality of complicated rating matrices to approximate ones is one useful strategy to mitigate the adverse impact of data sparsity [7,44,80]. For example, a latent factor model, matrix factorization, or singular value decomposition can accomplish this. We show that even a basic hybrid recommender system that simply combines user and item data can produce a better prediction than conventional systems.

Contrastive learning [100] can assist in addressing the problem of sparsity in recommendation systems. Sparsity is a situation in which there is insufficient user–item interaction data, making it difficult for standard models to anticipate accurately. Contrastive learning is a self-supervised learning method that seeks to acquire usable representations by differentiating between similar and dissimilar data points. In the context of recommendation systems, contrastive learning can be used to increase the model’s capacity to generalize and produce better suggestions by learning robust user and object representations, even when interaction data are limited.

According to the study in [100], contrastive learning outperforms conventional models in classification and exhibit enhanced accuracy through hyperparameter optimization and fine-tuning. The accuracy of a semi-supervised model with only 5% labeled data is 57.72% according to the results, whereas careful tuning in a supervised setting increases the accuracy to 88.70% [100].

#### 6.6.3. Alluvial Diagram

RAWGraphs is a high-quality open-source platform for developing unique data visualizations [101]. Figure 18 shows a graph generated with this tool to better comprehend data flow. This graph includes factors like document type, journal, and date of publication.

#### 6.6.4. Limitations and Biases

The deployment of hybrid recommender systems at scale confronts constraints such as high processing needs and latency concerns caused by complicated models. Additional issues include integrating varied data sources, retraining on a regular basis, and assuring interpretability. Cold-start issues, data scarcity, and algorithm scalability are all factors that influence performance. Balancing real-time customization with system response time and costs remains a challenge.

The biases we confront when reviewing abstracts and titles may impact our perception of relevance. Subconsciously, factors such as the authors’ reputation, the prestige of the journal, or even the authors’ names can influence our evaluation despite the precautions taken to prevent this from happening. However, it is critical to recognize that the topic of the abstract should not be the only factor influencing how we make choices.

We acknowledge that we were susceptible to biases during the manual screening process prior to using ASReview. One type of bias that impacts research papers is publication bias. Top-tier publications in almost all disciplines tend to publish papers with substantial findings, frequently accompanied by significant effect sizes. Using only the most prestigious publications may result in an overestimation of the effects in the field of interest. Lower-tier journals typically report smaller effect sizes in their publications. This search’s limitations include the authors’ exclusive use of academic databases for this investigation; therefore, they cannot ensure that all the relevant papers were located. A second method using artificial intelligence algorithms (ASReview) recommended the top articles based on relevancy to eliminate bias or misclassification. Finally, relevant items may have been excluded due to a lack of precision in the omission context of certain knowledge bases. While some articles clearly stated the context in which they were applied, many others did not. As a result, this study may not have considered other methodologies that are applicable to hybrid recommender systems.

The other biases can be summarized as follows:

Hybrid recommender systems, which combine two or more recommender techniques in order to improve the quality and effectiveness of tailored recommendations and applied methodology, may provide bias-related hazards and difficulties.

-**Data Bias:** Hybrid recommenders use data from several sources, each with inherent biases. For example, collaborative filtering algorithms rely on user–item interaction data, which can be skewed by popularity or suffer from the cold-start problem. Conversely, content-based approaches rely on item qualities, which may be prejudiced if the item descriptions are inadequate or skewed. Combining various data sources without considering their respective biases can result in biased suggestions.-**Algorithm Selection Bias:** In a hybrid system, various algorithms are used to handle different circumstances or specific jobs. The decision of which algorithm to apply for a specific user or environment may result in selection bias. If the system prefers one algorithm over another based on biased criteria, it may result in unfair or erroneous suggestions. For example, applying a specific algorithm just to certain user demographics may result in biased results.-**Combination Bias:** Hybrid systems usually integrate the outputs of several algorithms, which might result in bias. Different algorithms may have different biases, and, if the merging process is not carefully managed, it may exacerbate existing biases or create new ones.-**Feedback Loop Bias:** Hybrid recommenders, like other recommendation systems, are susceptible to feedback loop bias. A self-reinforcing loop can occur when the system’s recommendations influence user behavior, which is subsequently utilized to train the system. This bias can grow with time, particularly in hybrid systems with numerous algorithms contributing to the feedback loop. If the system fails to account for this prejudice, it may limit the diversity of the recommendations while reinforcing existing biases.-**Over-Specialization Bias:** Hybrid systems seek to increase performance by integrating methodologies; however, this can occasionally result in over-specialization. If the system is overly reliant on a single algorithm or data source, it may excel in some cases but underperform in others, resulting in biased suggestions. Balancing the contributions of various components in a hybrid system is critical for preventing this type of bias.-**Contextual Bias:** Hybrid recommenders frequently use contextual characteristics to generate individualized recommendations. However, biased or inadequate contextual information can result in biased outcomes. For example, using demographic data without addressing potential biases may result in suggestions that reinforce preconceptions.-**Evaluation Bias:** Evaluating the performance of hybrid recommenders can be difficult, and the selection of evaluation measures and test datasets may create bias. If the evaluation process favors some parts of the system’s performance, it may overlook or underestimate biases in other areas.

To reduce these dangers, researchers and developers should carefully design and assess hybrid recommender systems, taking into account fairness, diversity, and the potential biases of individual components and combinations. Implementing algorithms with fairness restrictions can help to balance recommendations across different user groups. Regular monitoring and user feedback can also assist in uncovering and correcting biases in real-world installations.

#### 6.6.5. Overfitting

The integration of some features in a recommendation system model can cause overfitting due to the absence of valuable and consistent information regarding the nature of the digital platforms under consideration [19]. Some additional contexts may not improve or perhaps have a negative impact on the model’s accuracy. However, this type of knowledge can be generalized and classified into more broad and intelligible categories.

### 6.7. Hybridization Stratégies (RQ2)

Several hybridization tactics have been investigated by researchers to improve the quality recommender system performance in the big data setting, where enormous volumes of user and item information are available. A few of the most important hybridization techniques used are as follows:

**Content–Collaborative Hybridization:** Combining collaborative filtering, which makes use of past preferences and user–item interactions, with content-based filtering, which makes use of item attributes and user profiles, is known as content–collaborative hybridization. Combining collaborative- and content-based signals enables this hybrid technique to deliver suggestions that are more thorough and precise. The research in [71] offers an ontology-based model that combines multi-level k-means, rough set, and Bayesian network to beat SVM, DT, and RF with the lowest log error loss and 98% accuracy.

**Deep-Learning-Based Hybrid Recommenders:** New developments in deep learning methods, like neural networks and embeddings, have made it possible to create hybrid recommender systems that efficiently manage complicated large-scale data. Recommendations from deep-learning-based models are more precise and tailored because they are able to identify complex patterns and linkages in user–item interactions. The study in [70] solves various research challenges by creating a CNN-based no-reference video quality assessment for gaming footage that is impacted by compression artifacts.

**Hybrid Matrix Factorization:** By adding more data, hybrid matrix factorization approaches build upon the foundation of standard matrix factorization techniques. This can involve adding hybrid regularization words, user or item traits, or side information. The method is able to capture more intricate associations and enhance the quality of recommendations by including hybridization in the factorization process. The study in [31] introduces a hybrid content-based and neighborhood-based recommender model that uses a new similarity measure. It achieves accuracy similar to innovative item-oriented and matrix factorization models while running at least twice as fast.

**Demographic–Collaborative Hybridization:** Combining collaborative filtering algorithms with user demographic data, such as age, gender, location, or socioeconomic status, this hybrid paradigm, which combines collaborative patterns with user-specific features, can improve personalization and tackle the cold-start issue.

The paper [33] presents a hybrid strategy that combines collaborative filtering and demographic recommendation systems, utilizing data mining, artificial neural networks, and fuzzy techniques.

**Knowledge-Based Hybridization:** It enhances the recommender system’s comprehension of user preferences and item linkages by integrating domain-specific knowledge, rules, or ontologies. With this hybrid method, more context and explanation may be provided. In the article [19], the author created a Music Information Knowledge Graph (MKG) that contains user-track interaction pairs, track content attributes, and artist context elements.

### 6.8. Datasets (RQ3)

In response to RQ3, we followed the available datasets that the writers used to evaluate their hybrid recommendation systems (HRSs). These databases enable the scientific community to reproduce studies and validate or enhance their procedures. Out of the fifty-two studies, forty-eight used at least one dataset, whereas three did not. Figure 19 depicts the datasets used and their frequencies among the studies.

The findings show a heterogeneous sector of dataset utilization, with a few web and survey datasets dominating the research landscape while also including less prevalent datasets. This distribution might provide information pertaining to the research trends and preferences regarding the topic.

**Dataset Distribution:** The table depicts the distribution of studies among different datasets used for evaluation. The most common datasets are “web dataset” and “survey data”, accounting for 26% of all the research.

**Concentration of Studies:** The results show that the studies are concentrated on specific datasets. The top three datasets (“web dataset”, “survey data”, and “social media data”) account for more than half of all the studies, indicating that the research community prioritizes these types of datasets.

**Diversity of Datasets:** While the most prevalent datasets dominate the distribution, the table also includes less common datasets, such as “Instructional materials”, “Qualitative Data”, “Synthetic dataset”, and “Clinical Data”, which account for 2–4% of all the studies. This indicates a degree of diversity in the datasets utilized for study.

**Balanced Representation:** The distribution appears to be somewhat balanced, with no single dataset accounting for an overwhelming majority (the largest percentage is 26% for both “web dataset” and “Survey data”). This shows a healthy diversity of datasets used in the investigations.

**Missing or Unspecified Data:** The 6% of research labeled as “NA” (not available) indicates that a minor amount of data may be missing or undefined in the source material.

### 6.9. Experimental Outcomes (RQ4)

Hybrid recommender systems frequently seek to use the capabilities of various recommendation methodologies (e.g., content-based, collaborative filtering, and demographic-based) in order to provide more accurate and personalized recommendations to users. Combining various algorithms can improve recommendation performance, as assessed by measures such as precision, recall, F1-score, or normalized discounted cumulative gain. The study in [31] provides a hybrid recommendation system that blends content-based and neighborhood-based algorithms to increase accuracy and speed. It employs novel approaches to improving item-level similarity measures in collaborative filtering algorithms (see Table A1). The work employs genomic tags and aims to outperform the traditional collaborative filtering methods in terms of accuracy and speed. The experiment results indicate that it is more precise and faster than ‘pure’ collaborative filtering techniques.

The study in [61] incorporates both conventional and additional aspects pertaining to pandemic, environment, digital technology, and information systems; the study offers a thorough methodology for assessing airline service quality.

Ref. [71] utilized an ensemble approach consisting of three techniques: clustering, rough set, and Bayesian network. The strategy was divided into four phases: clustering, knowledge discovery, probabilistic network design, and model evaluation. Based on experimental data, this model outperformed other models like DT, RF, and SVM, with an accuracy of 98.36% (several further results are included in Table A1 in Appendix A.

### 6.10. Methodologies and Recommended Techniques (RQ5)

According to Table A1, Column 2 in Appendix A, the proposed technique for hybrid recommendation systems typically includes the following important steps:-**Data Collection:** Gather data from various sources, including user behavior logs, questionnaires, interviews, item metadata, and user profiles.-**Feature Engineering:** Relevant qualities that influenced proposals were identified and selected. To increase model performance, more features were developed using existing data. Categorical variables were encoded utilizing techniques like one-hot encoding and embedding [60,94].-**Employ the Strengths of Different Methods:** Hybrid systems combine the benefits of several recommendation techniques, such as those based on content, collaborative filtering, and demographic information, to take advantage of their respective capabilities and provide more precise and personalized recommendations.-**Experiment and Evaluate Performance:** Experiments are carried out to evaluate the performance of hybrid systems regarding individual recommendation strategies. The increases in recommendation accuracy are evaluated using metrics like as precision, recall, F1-score, and normalized discounted cumulative gain.-**Address Individual Technique Limits:** Hybrid systems are intended to overcome the limits of individual recommendation approaches, such as the cold-start problem or the inclination to propose primarily popular goods. The experiments show increased coverage of long-tail items and more diverse recommendations according to users’ unique interests.-**Analyze Efficiency and Scalability:** The study compares the computational efficiency, memory utilization, and scalability of hybrid strategies to individual recommendation approaches. The experiments evaluate hybrid systems’ processing times, memory footprints, and applicability for real-world big data applications.-**Assess Customer Experience and Satisfaction:** Experiments are carried out to assess the influence of hybrid systems on user experience, engagement, loyalty, and overall satisfaction. The efficacy of the hybrid techniques is measured by analyzing user input, engagement metrics, and satisfaction levels.-**Appreciate Hybridization:** Experiments are intended to highlight any trade-offs connected with hybridization, such as the effect on model transparency, interpretability, or the complexity of the recommendation process. These findings can help to inform future system design decisions and the selection of appropriate hybridization strategies.-**Identify Optimal hybridized Strategies:** Experiments are performed to determine the best ways to combine several recommendation approaches, such as weighted hybrid, switching hybrid, feature augmentation, and meta-level hybrid. The study provides practitioners with guidance for selecting and implementing hybrid approaches depending on the data characteristics and intended recommendation performance.

### 6.11. Potential Future Research Directions (RQ6)

The last study question concerns the future job prospects and directions. Our results are reported in Table 11 and briefly discussed below:

For the study in [31], employing a recommendation system as an integrated Movie Sales Recommendation Engine, future work will focus on enhancing movie representations and integrating matrix factorization techniques for increased accuracy.

The authors of the study in [33] on a hybrid model in social networks recommendation system architecture development will evaluate their techniques on more social networks and investigate the possibility of combining them with genetic algorithms for better outcomes.

The authors of the study in [36] examined courseware and open educational resources with an emphasis on quality. One of their future objectives is to automate processes related to the creation of an effective and personalized adaptive recommendation system. Future plans call for automating several framework operations to enhance flexibility and recommendations. Developing an excellent adaptive recommender system that is tailored to users’ learning needs is the ultimate objective.

One of the goals for the future is task automation for the development of a personalized and effective adaptive recommender system. The plans for the future include automating some framework activities to enhance recommendations and flexibility. Creating a superior, personalized adaptive recommender system for users’ learning needs is the ultimate objective.

One of the primary goals of the project management system study [75] is to raise the general standard of Jakarta’s municipal parks; subsequent studies could concentrate on raising the administration and management of Jakarta’s parks, as well as raising the administration of construction projects, especially in the pre-construction stage.

The authors of [81] underline the importance of assessing and monitoring societal perceptions of enhanced individuals. They contend that understanding these perspectives is critical for guiding the development and use of future augmentation technologies.

According to the article [78] on financial modeling techniques, future gains can be achieved by adjusting reimbursement structures and implementing quality-based incentives.

Building on our previous responses, we conducted in-depth evaluations of each individual study to correctly address research questions RQ2 to RQ6. The goal was to document the technical methods, algorithms, approaches, and findings utilized in developing hybrid recommender systems as described in the literature. Table A1 in Appendix A shows a summary of the employed strategy, the dataset used, the objectives, and the results. As presented in Appendix A, the hybrid recommendation systems used a variety of approaches to improve accuracy, coverage, and user experience. The experiments found that hybrid systems outperformed individual techniques in terms of precision, recall, and diversity. The hybrid techniques also demonstrated higher efficiency and scalability in large-scale applications. The evaluations of user feedback and interaction revealed that personalized, relevant recommendations increased satisfaction. The experiments revealed trade-offs in hybridization strategies and helped to identify the optimal procedures for specific applications.

#### Future Research

Given the findings of this study, we see potential for further research in context-sensitive systems and hybridization techniques. To efficiently design CARS, the following tools and approaches could be used:**Context-Aware Recommendation Systems (CARSs)** enhance traditional recommendation models by integrating contextual factors, such as location, time, or environmental conditions, into the recommendation process, developing techniques for gathering contextual data, such as user behavior analytics or environmental sensors, and designing algorithms that include contextual information in the recommendation process. Unlike conventional systems that predict ratings based only on user–item interactions (F:User×Item→Rating), CARSs expand the prediction function to include context (F:User×Item×Context→Rating), adding a third dimension. This added complexity makes the recommendations more relevant by aligning them with situational user needs, although it also increases the computational demands. A clear understanding of “context”, defined as any information shaping the user interactions with the system, is essential for effectively designing these systems. To efficiently design a CARS, the following tools and approaches could be used:
**Machine Learning Frameworks:** Use machine learning frameworks such as TensorFlow or PyTorch to create prediction models based on contextual information. These frameworks provide strong libraries for developing and training machine learning models, enabling the integration of complicated characteristics such as context in addition to user and object data.**Dataset, Model, and Evaluation:** Creating a contextual dataset, creating a reinforcement learning model, and using performance measures to evaluate adaptation.**Contextual Bandits:** Use contextual bandit algorithms to dynamically adjust recommendations based on real-time circumstances. These algorithms strike a balance between exploration and exploitation by determining which recommendations function best in various contextual settings, allowing the system to deliver tailored ideas that adapt as user behaviors and contexts change.**User Studies:** Conduct user research to determine the effectiveness of context-aware recommendations. Gathering qualitative input from consumers allows us to measure how effectively the recommendations suit their needs and preferences in various scenarios. This approach may include surveys, interviews, or A/B testing to measure user happiness and engagement with contextual features.**Hybridization:** In machine learning, hybridization is the process of merging multiple algorithms or models to improve predicted accuracy, resilience, and flexibility by utilizing their strengths while mitigating individual limitations. Hybridization in recommendation systems frequently employs ensemble learning techniques such as stacking and meta-learning to combine collaborative- and content-based filtering methods. This method enhances recommendation accuracy by modifying model weights in response to user interactions. Scikit-learn and PyTorch are tools that help to apply these concepts, making it easier to experiment and enhance hybrid systems across a wide range of applications, including recommendation engines, identifying fraud, and natural language processing. To supplement the conversation, we will provide a more detailed examination regarding how these concerns could be investigated:
**Frameworks for Hybrid Systems:** Use libraries that support hybrid recommendation algorithms, such as Surprise or Apache Mahout.**A/B Testing:** Use A/B testing techniques to compare the performance of hybrid models to standard approaches.**Data Fusion Techniques:** Explore data fusion approaches to successfully merge multiple sources of data, hence improving the quality of recommendations.

The gaps in the literature require more exploration. Addressing these deficiencies is critical to improving the scalability, accuracy, and ethical issues of hybrid recommender systems:

**Scalability Challenges:** To manage enormous datasets efficiently, scalable techniques are required.

**Integration of Advanced AI techniques:** Investigating how deep reinforcement learning and generative models might improve recommendation accuracy.

**Data Privacy and Ethical Considerations:** Creating techniques for implementing privacy-preserving procedures while maintaining the quality of suggestions.

**Experience and Engagement Metrics:** Focusing on the importance of increasing user happiness and trust in the advice provided.

## 7. Conclusions

This article provided a comprehensive survey and assessment, as well as an extended organized taxonomy, for the most recent, ever-increasingly efficient hybrid recommendation system models used in both academia and industry, with successful applications in fields such as e-commerce, music, and geographic location services. In this work, we employed an open-source system that uses machine learning to efficiently filter and categorize large amounts of textual data, which sped up the document selection process. Using this approach in conjunction with the traditional methods, we discovered 52 key publications from conference proceedings and journals on hybrid recommender systems. Our goal was to highlight the most relevant concerns addressed by these studies in order to make more informed suggestions. We also studied the machine learning and data mining approaches they employ, the recommendation strategies they merge, the hybridization classes they adhere to, the application domains and datasets, the evaluation procedure, and potential future work paths. A significant portion of the research we examined (more than 75%) was published during the last three years, demonstrating a noticeable and growing interest in hybrid recommender systems (HRSs). This work emphasizes the need for further research into context-sensitive systems and hybridization tactics in context-aware recommendation systems (CARSs). By incorporating contextual aspects, a CARS improves the traditional models, improving their relevance while increasing the processing demands. Machine learning frameworks, contextual bandits, and user studies are key tools for assessing effectiveness. Furthermore, hybridization combines algorithms to improve accuracy, with an emphasis on using frameworks such as Apache Mahout, while also addressing scalability, ethical concerns, and user engagement metrics in future studies. Furthermore, our outcomes indicate that using larger datasets and hybrid parallel algorithms may be a viable way to handle scalability issues and improve recommendation quality in the age of big data. Another intriguing area for future research is the use of hybrid recommendation systems to create cross-domain recommenders or lower the computational complexity of the existing approaches.

## Figures and Tables

**Figure 1 jimaging-11-00012-f001:**
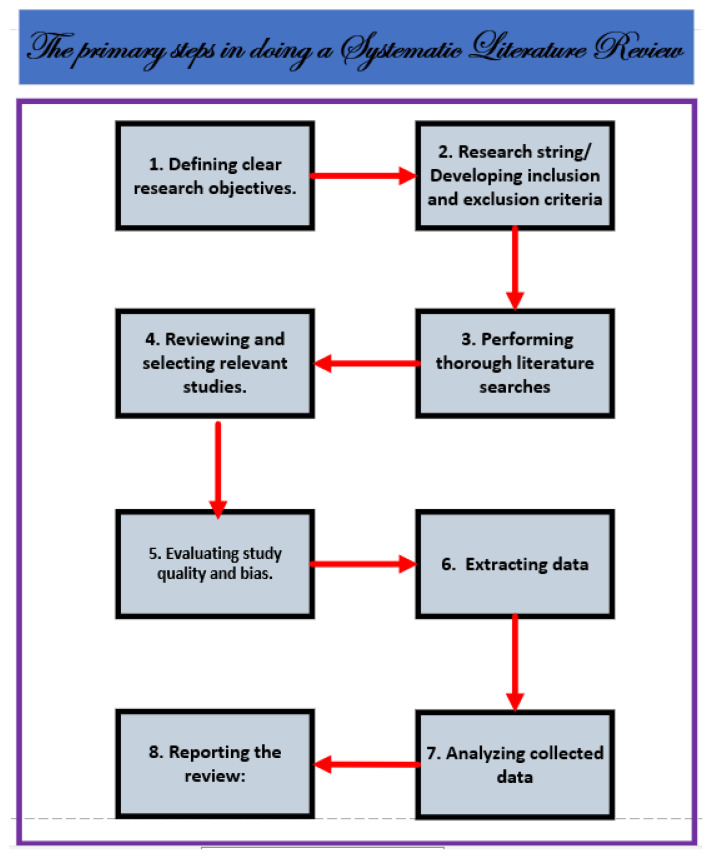
The basic steps in conducting a systematic literature review.

**Figure 2 jimaging-11-00012-f002:**
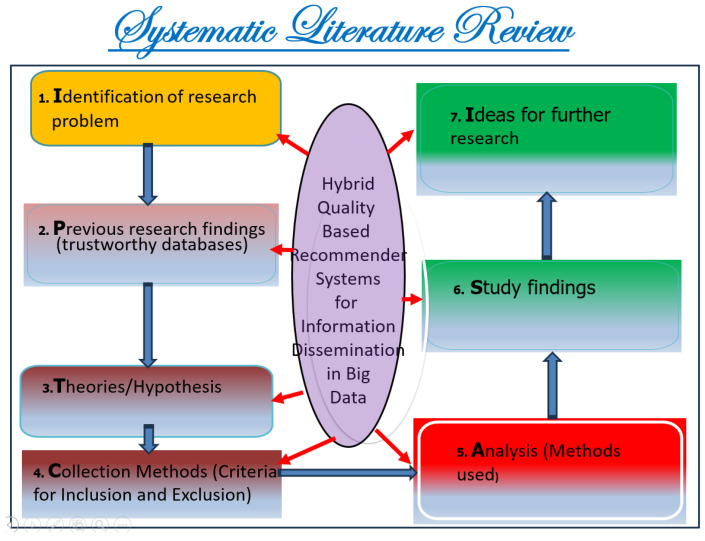
Literature review process template.

**Figure 3 jimaging-11-00012-f003:**
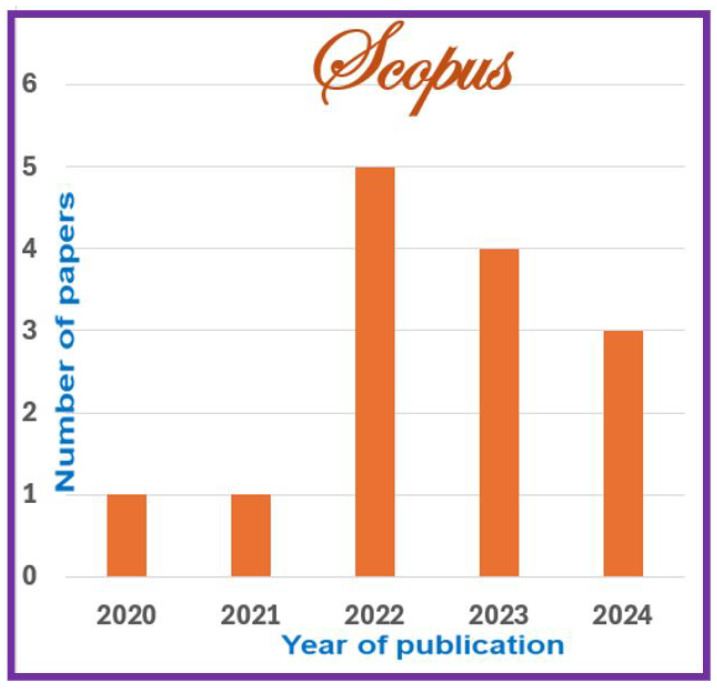
Scopus: number of articles published in the study area from 2020 to 2024.

**Figure 4 jimaging-11-00012-f004:**
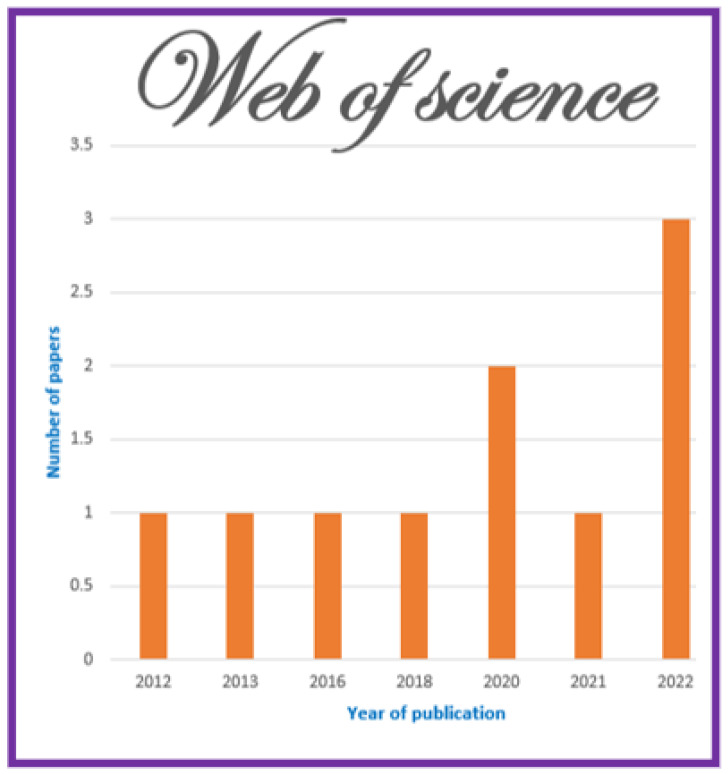
Web of Science: number of articles published in the study area.

**Figure 5 jimaging-11-00012-f005:**
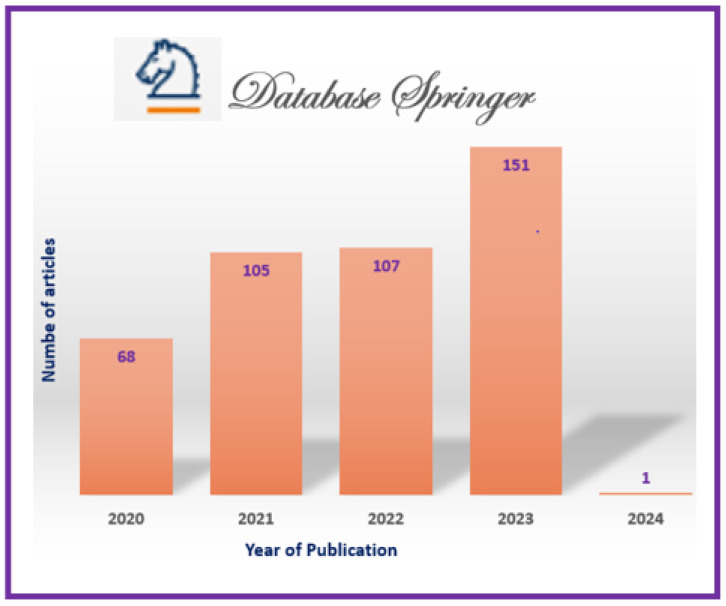
Springer: number of articles published in the study area from 2020 to 2024 without preview-only content.

**Figure 6 jimaging-11-00012-f006:**
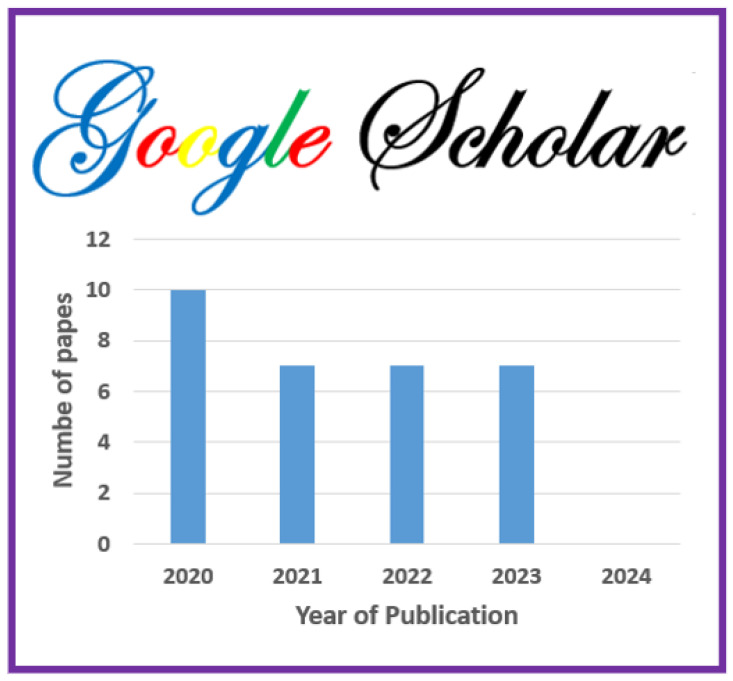
Google Scholar: number of articles published in the study area from 2020 to 2024.

**Figure 7 jimaging-11-00012-f007:**
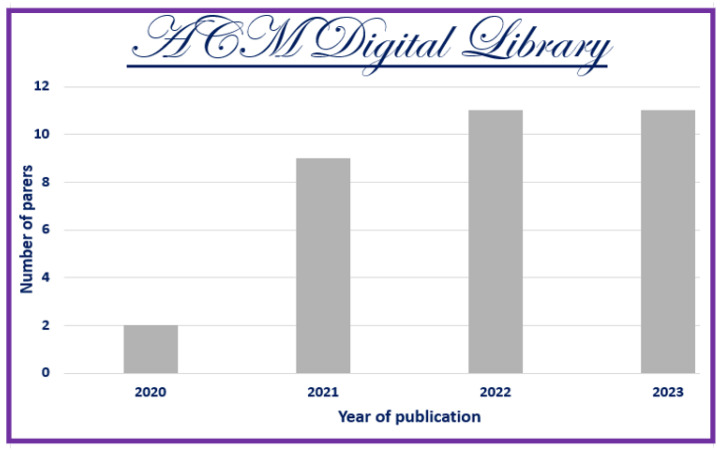
ACM Digital Library: number of articles published in the study area.

**Figure 8 jimaging-11-00012-f008:**
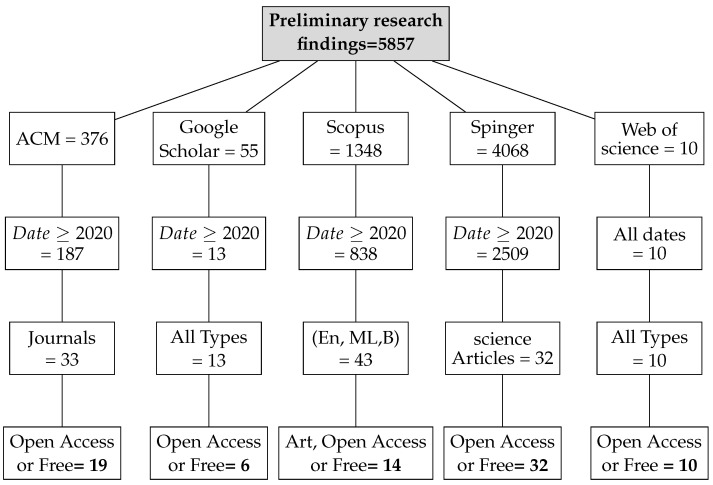
Data selection methods.

**Figure 9 jimaging-11-00012-f009:**
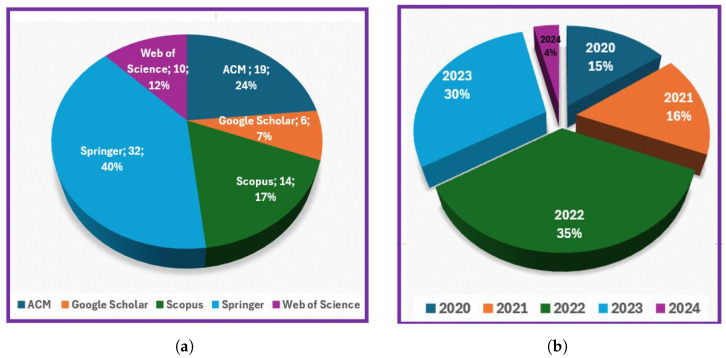
Article counts for the distribution of academic paper databases. (**a**) Ratio of articles vs. database. (**b**) Academic paper database spread.

**Figure 10 jimaging-11-00012-f010:**
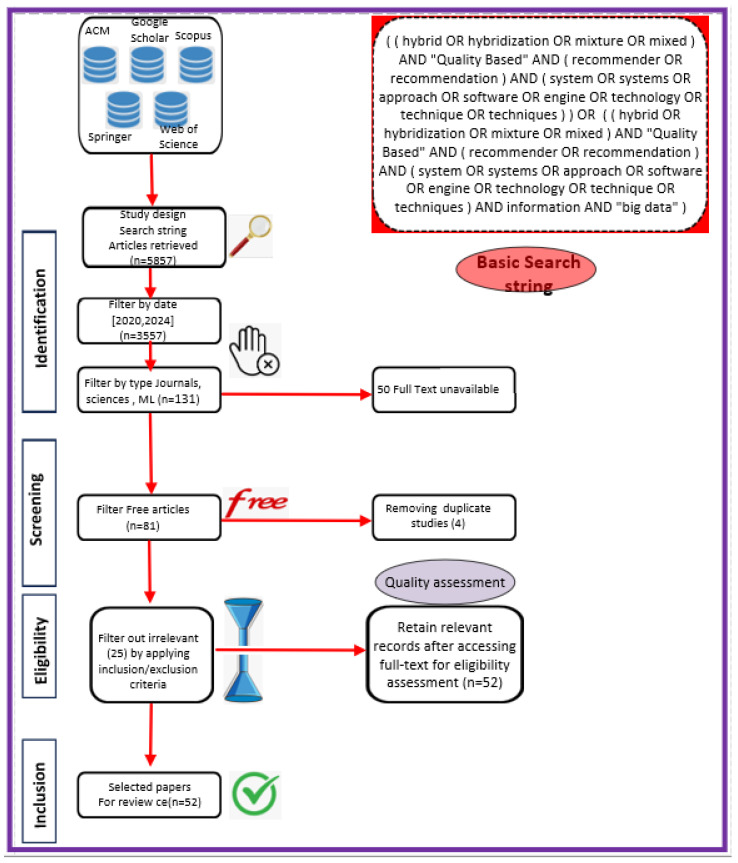
The research approach employed: diagram of the PRISMA process for inclusion and exclusion.

**Figure 11 jimaging-11-00012-f011:**
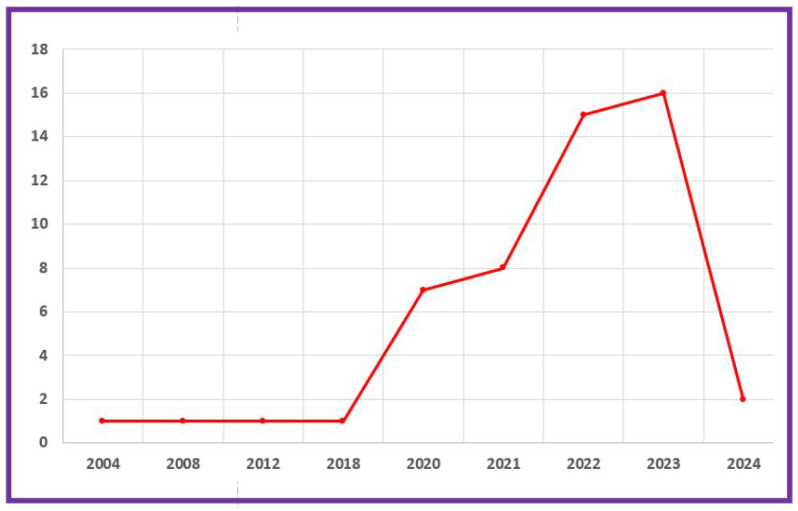
Spread of research based on the publication year of chosen papers.

**Figure 12 jimaging-11-00012-f012:**
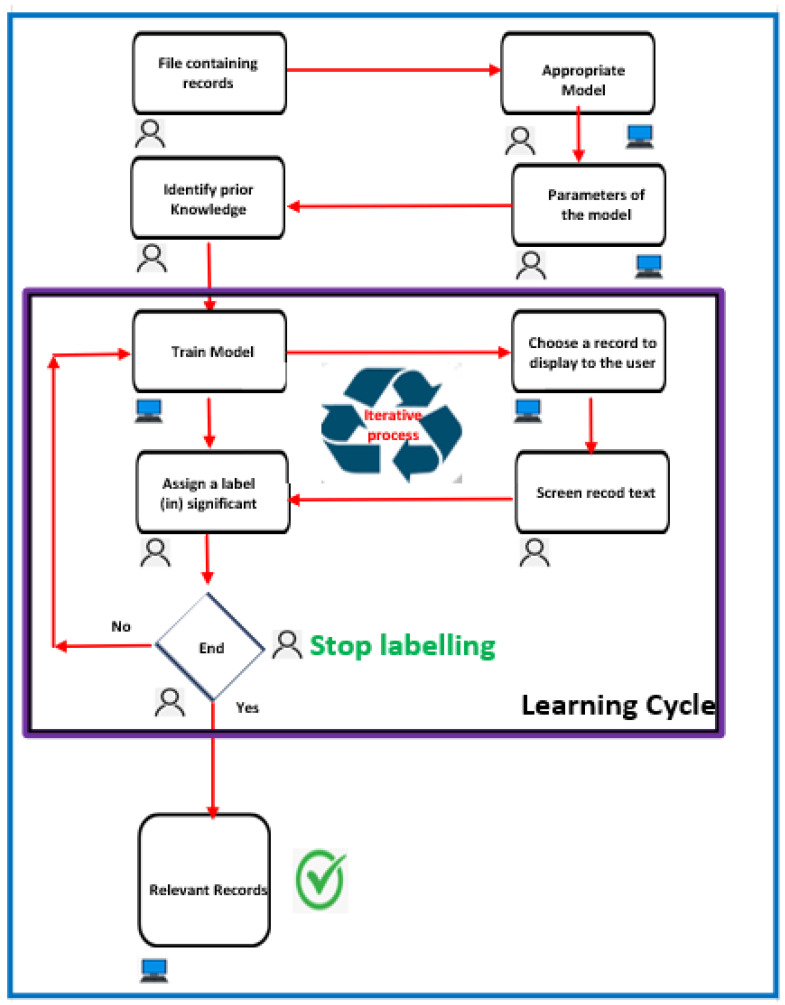
Machine-learning-based ASReview pipeline. Graphic icons denote actions performed by human or computer.

**Figure 13 jimaging-11-00012-f013:**
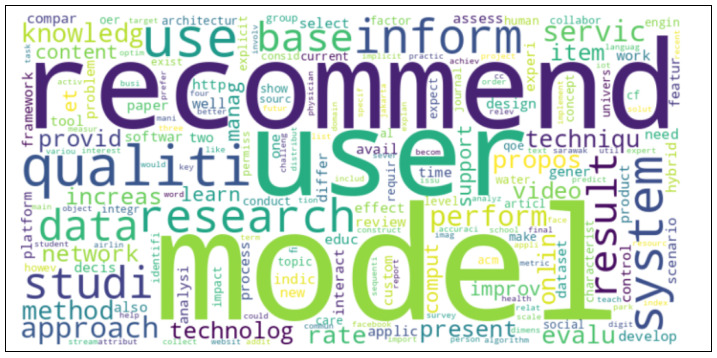
Top 1000 abstract words.

**Figure 14 jimaging-11-00012-f014:**
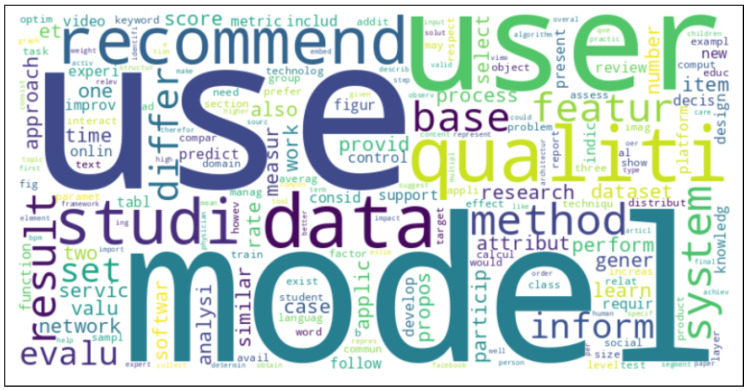
Top 1000 words in whole papers.

**Figure 15 jimaging-11-00012-f015:**
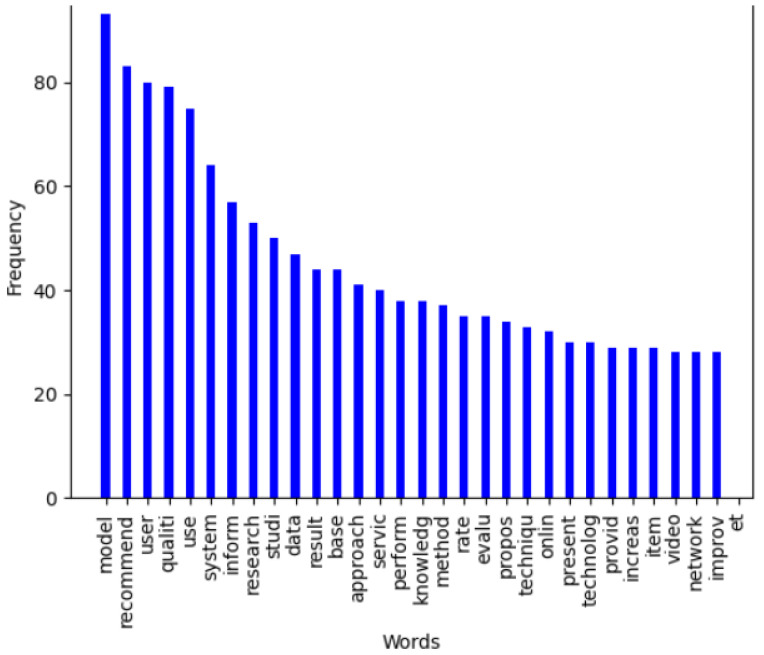
Word frequency in abstracts (top 30).

**Figure 16 jimaging-11-00012-f016:**
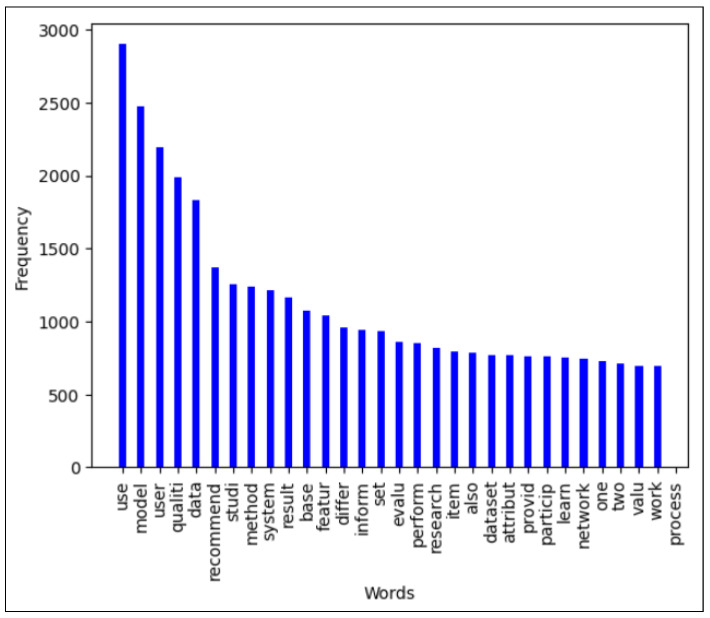
Word frequency: top 1000 words in whole papers.

**Figure 17 jimaging-11-00012-f017:**
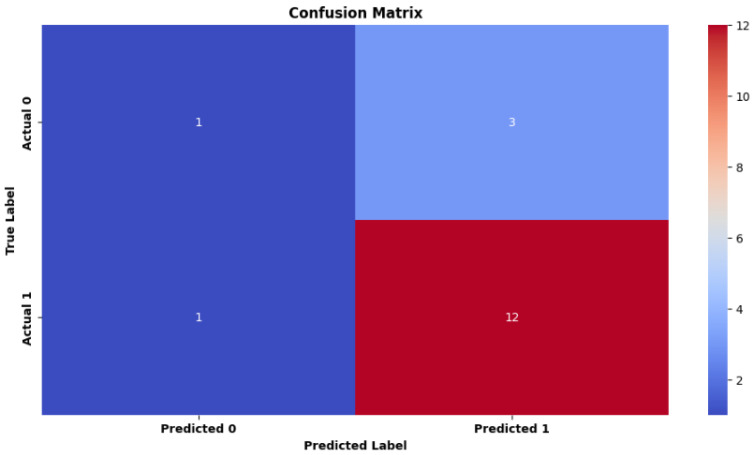
Confusion matrix for the articles selected for the study.

**Figure 18 jimaging-11-00012-f018:**
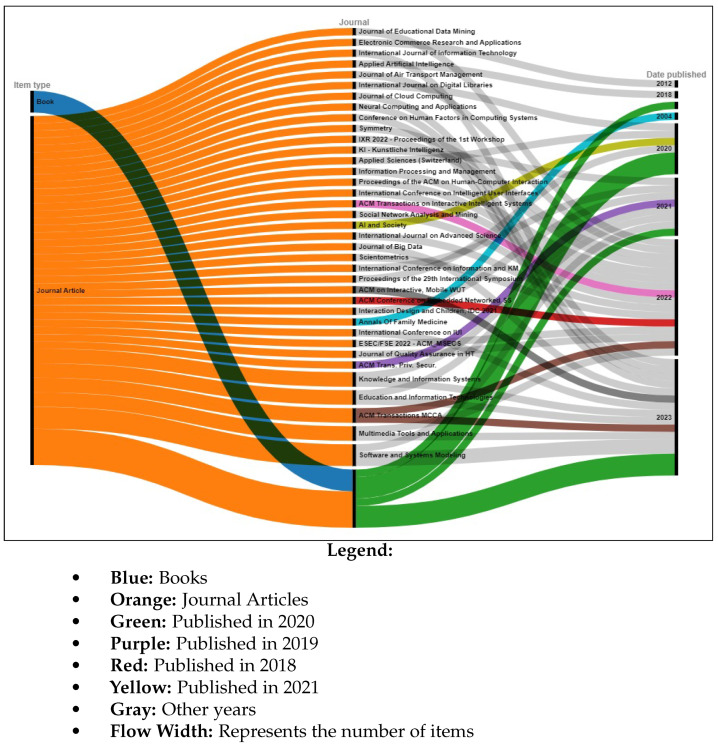
Multicategorical article analysis with a complete color-coded legend.

**Figure 19 jimaging-11-00012-f019:**
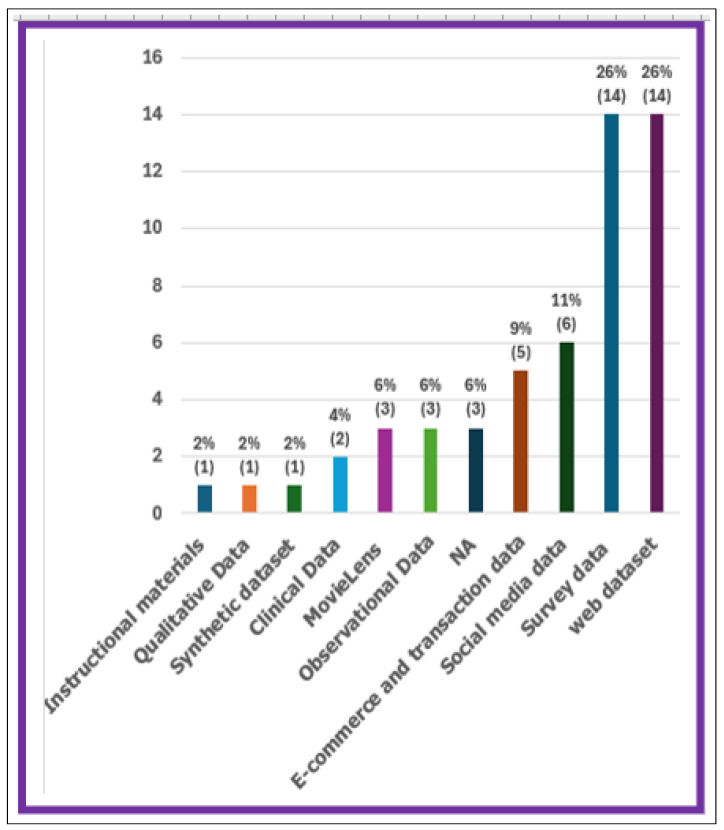
Trends in using assessment datasets for recommender system research.

**Table 1 jimaging-11-00012-t001:** SLR: research questions.

Research Questions	Motivation and Projected Results
**RQ1.** What are the relevant studies on hybrid recommenders, and how do hybridization techniques solve specific difficulties such as cold-start, novelty, diversity, and user satisfaction?	Identifying challenges connected to recommendation systems (Data Sparsity, Model Bias, Overfitting, and Dimensionality Reduction).
**RQ2.** What are the various hybridization strategies that have been employed to increase the performance of quality recommender systems in the context of big data?	Address the issues that come with developing effective quality recommender systems in a setting of massive amounts of data.
**RQ3.** What types of data sources have been used to evaluate the techniques in recently published hybrid recommendation systems?	To pinpoint contributions closely associated with using recommendation systems for proposing housing alternatives.
**RQ4.** What experimental outcomes are generated when hybrid recommender techniques are used?	To increase the overall performance and efficacy of recommendation systems, especially in large-scale, complex data contexts.
**RQ5.** What is the suggested methodology in hybrid recommendation systems?	Identify the proposed methods in hybrid quality-based recommendation systems.
**RQ6.** What are the most promising future research directions?	Determine potential research directions for improving hybrid quality-based recommendation systems.

**Table 2 jimaging-11-00012-t002:** Keywords and synonyms.

Keyword	Synonyms
Hybrid	Hybridization, Mixture, Mixed
System	Systems, Approach, Software, Engine, Technology, Technique, Techniques
Recommender	Recommendation

**Table 3 jimaging-11-00012-t003:** Basic query.

* Basic Search String
((hybrid OR hybridization OR mixture OR mixed) AND “Quality Based” AND (recommender OR recommendation) AND (system OR systems OR approach OR software OR engine OR technology OR technique OR techniques)) OR ((hybrid OR hybridization OR mixture OR mixed) AND “Quality Based” AND (recommender OR recommendation) AND (system OR systems OR approach OR software OR engine OR technology OR technique OR techniques) AND information AND “big data”)

**Table 4 jimaging-11-00012-t004:** Scopus: the search string keywords with filters.

* Scopus: Advanced Search Keywords
((hybrid OR hybridization OR mixture OR mixed) AND “Quality Based” AND (recommender OR recommendation) AND (system OR systems OR approach OR software OR engine OR technology OR technique OR techniques)) OR ((hybrid OR hybridization OR mixture OR mixed) AND “Quality Based” AND (recommender OR recommendation) AND (system OR systems OR approach OR software OR engine OR technology OR technique OR techniques) AND information AND “big data”) AND PUBYEAR > 2019 AND PUBYEAR < 2025 AND (LIMIT-TO (OA, “all”)) AND (LIMIT-TO (SUBJAREA, “ENGI”) OR LIMIT-TO (SUBJAREA, “COMP”) OR LIMIT-TO (SUBJAREA, “BUSI”)) AND (LIMIT-TO (LANGUAGE, “English”)) AND (LIMIT-TO (EXACTKEYWORD, “Machine Learning”)) AND (LIMIT-TO (DOCTYPE, “ar”))

**Table 5 jimaging-11-00012-t005:** Google Scholar: the search string keywords with filters.

* Google Scholar: Advanced Search Keywords
((hybrid OR hybridization OR mixture OR mixed) AND (recommender OR recommendation) AND (system OR systems OR approach OR software OR engine OR technology OR technique OR techniques)) OR ((hybrid OR hybridization OR mixture OR mixed) AND (recommender OR recommendation) AND (system OR systems OR approach OR software OR engine OR technology OR technique OR techniques) AND information)

**Table 6 jimaging-11-00012-t006:** Dissemination of papers sourced from academic databases.

Database Source	Retrieval	Preliminary Removal	Second-Level Selection
ACM	376	187	19
Google Scholar	55	13	6
Scopus	1348	838	14
Springer	4068	2509	32
Web of Science	10	10	10
Total			

**Table 7 jimaging-11-00012-t007:** Form for extracting data.

Extracted Data	Explanation	RQ
Title	The name of the article	RQ1
Authors	-	-
Description	Brief overview of the paper’s content	-
Publication year		RQ1
Source	Source of digital library access	RQ3
Publisher	-	-
Application domain	Application domain of the study	-
Approach	Methodology employed	RQ2, RQ5
Contribution	Research work’s significance	-
Evaluation methodology	Approach to evaluating the recommender system	RQ6
Dataset	Data repository	RQ4
Experiment	Explanation of the experiment	RQ4
Future work	Proposed future research areas	RQ6

**Table 8 jimaging-11-00012-t008:** PRISMA 2020 Checklist.

Section/Topic	#	Item	Page Where the Item Is Reported
**TITLE**			
Title	1	This report describes a systematic review conducted in accordance with PRISMA guidelines. The goal of this review was to summarize the evidence on hybrid recommender systems.	1
**ABSTRACT**			
Abstract	2	Systematic reviews use rigorous methodologies to provide a thorough assessment of relevant studies while combining existing knowledge on specific issues. Following the standards in the Cochrane Handbook, Kitchenham, and Charters ensures transparency and quality. This paper also evaluates hybrid recommendation systems, emphasizing their expanding importance and potential future research avenues, such as incorporating contextual information and enhancing scalability with sophisticated algorithms. A strong emphasis is placed on the effectiveness of machine learning in filtering relevant material on these systems.	1, 8, 19, 36
**INTRODUCTION**			
Rationale	3	The review of hybrid recommendation systems discusses their increasing importance in providing individualized user experiences while overcoming the constraints of older methods. It seeks to identify best practices, emerging trends, and future research directions that will improve the effectiveness and flexibility of these systems.	2, 3, 4, 45–66
Objectives	4	The paper attempts to consolidate existing knowledge on hybrid recommendation systems, identify best practices, and assess emerging machine learning trends. It also aims to identify research gaps, present a consistent evaluation system, and guide practical applications to improve user experiences.	3, 4, 45–66
**METHODS**			
Eligibility Criteria	5	**IC1:** Papers offering hybrid quality-based recommender systems, algorithms, and techniques in the context of big data. **IC2:** Papers from conferences and journals published between 2020 and 2024. **IC3:** The paper incorporates search-relevant keywords within its title or abstract. **IC4:** The paper addresses hybrid recommendation systems. **IC5:** The paper addresses at least one problem of recommendation or proposes at least one technique of hybridization. **EC1:** The publication date is earlier than 2020. **EC2:** The paper is written in a language other than English. **EC3:** The paper is a short article, a standard, a poster, an editorial, or a tutorial. **EC4:** The title, abstract, and keywords are not relevant to the research topic. **EC5:** The paper does not discuss hybrid recommendation systems.	16
Information Sources	6	Using specific search keywords, we searched Scopus, ACM, Web of Science, Springer, and Google Scholar.	17, 18
Search Strategy	7	Scopus’ search method included employing specific terms such as “Hybrid Quality Based Recommender Systems”, “Information”, and “Big Data”, paired with Boolean operators. The search was restricted to publications published from 2020 to early 2024, with emphasis on relevant subject areas and document types.	12, 14, 18
Selection Process	8	Two independent reviewers first choose titles and abstracts, then analyze the complete text of the selected research. Any disagreements were handled through consensus on articles that were not retained by the two authors. An additional perspective was gained utilizing the ASReview tool to ensure a comprehensive and impartial selection process.	10–20, 28
Data Collection Process	9	We ensured validity by conducting a double extraction process by independent reviewers after going through the entire text of the included articles to methodically extract and summarize the data in a standardized table format to make comparisons easier. Choosing the pertinent data points, constructing and testing the extraction table, checking the gathered data for mistakes, and, if required, updating and pilot testing the approach are all part of this process.	17–20
Data Items	10	Data extraction was utilized to look for factors such as the study’s subject, strategy, sample size, demographic characteristics, objectives, data gathering techniques, and outcomes. These factors enable a comprehensive examination and comparison of studies.	45–66
Risk of Bias Assessment	11	We evaluated the risk of bias using the Cochrane Risk of Bias Tool, which included independent reviews by two reviewers and an open-source application. Discrepancies were resolved collectively, and the outcomes were thoroughly documented for analysis.	37–38
Effect Measures	12	Commonly Used Principal Summary Measures (Precision, recall, and F1-score).	35–36, 41
Synthesis of Results	13	A systematic literature study for hybrid recommender systems begins with data extraction, which is organized and standardized, followed by method categorization and statistical evaluations of performance measures. Meta-analysis, visualization tools, and thematic synthesis are used to combine and understand findings from multiple studies.	24–45
Reporting Biases	14	Describe any methods used to assess the risk of bias due to selective reporting.	38, 39
Certainty Assessment	15	The assessment of evidence certainty, which takes into account study quality, bias risk, and consistency, guarantees solid results and conformity to quality and transparency requirements.	18, 19, 32, 35
**RESULTS**			
Study Selection	16	Present the number of studies screened, assessed, and included, with reasons for exclusions.	25, 35, 40
Study Characteristics	17	For each included study, present characteristics (e.g., participants and interventions).	45–66
Risk of Bias in Studies	18	Present risk of bias judgments for each included study.	38, 39
Results of Individual Studies	19	For all outcomes considered, present the results of each study.	37–38
Synthesis of Results	20	Present results of syntheses (e.g., meta-analyses), including confidence intervals.	45–66
Reporting Biases	21	Report on the presence of any selective reporting.	
Certainty of Evidence	22	Present an assessment of the certainty (e.g., GRADE).	18, 19, 32, 35
**DISCUSSION**			
Summary of Evidence	23	Summarize the main findings, including the strength of evidence.	4, 6, 7, 36, 38
Limitations	24	Discuss limitations of the evidence and the review process.	15, 18, 19, 38, 39
Conclusions	25	Provide a general interpretation of the results in the context of other evidence.	10, 12, 37, 38, 42, 44
**FUNDING**			
Funding	26	Describe sources of funding and other support for the review.	Not Available

**Table 9 jimaging-11-00012-t009:** Main selection of papers identified by categories, journals, and publishers.

Primary	Author	Publisher	Year	Journal
	**Category**			
Collaborative Filtering	[60]	Springer Nature	2023	Int. Jrnl. of Tech
	[30]	Elsevier BV	2012	Elect. Commerce Research
	[16]	Google Scholar	N/A	Google Scholar
	[19]	Johannes Kepler	2021	N/A
Quality	[15]	Springer Berlin	2023	Jrnl Cloud Comp.
	[36]	Computer Science	2013	Comp. Col. Int
	[18]	Appl. Sci.	2020	Applied Sciences
	[29]	ACM	2022	Jrnl. Edu. D.Mng.
Content-based Based Filtering	[60]	Springer Nature	2023	Int. Jrnl. of Tech
	[31]	Springer, Cham	2020	Adv.Net. Inf. Systems
	[30]	Elsevier BV	2012	Elect. Commerce Research
	[16]	Google Scholar	N/A	Google Scholar
	[19]	Johannes Kepler	2021	N/A
Hybrid filtering	[14]	Journal Of King Saud University	2022	Journal Of King Saud University
	[29]	ACM	2022	Jrnl. Edu. D.Mng.
	[30]	Elsevier BV	2012	Elect. Commerce Research
	[35]	Elsevier Ltd	2022	Inf. Proc. and Mngt
	[33]	AI and Society	2020	AI and Society
	[31]	Springer, Cham	2020	Adv.Net. Inf. Systems
	[18]	Appl. Sci.	2020	Applied Sciences
	[36]	Computer Science	2013	Comp. Col. Int
	[34]	Springer Int. Publish.	2023	Journal of Big Data
	[13]	Taylor and Francis	2018	Applied AI
	[16]	Google Scholar	N/A	Google Scholar
	[32]	Springer	2020	Int. Jrnl on D.Lib.
	[15]	Springer	2021	Knowledge and Inf. Syst.
Other filtering	[61]	[62,63,64,65,66,67]	[68]	[69,70,71,72]
	[73]	[74,75,76,77,78,79]	[80]	[81,82,83,84]
	[85]	[86,87,88,89,90,91]	[92]	[93,94,95,96]

*Legend: N/A = Not available.*

**Table 10 jimaging-11-00012-t010:** Questions to evaluate the studies’ quality.

N#	Quality Question	Weight
1	Has the study looked over the relevant research for the issues?	1
2	Did the study adequately describe the issue it is trying to solve?	1
3	Was an experimental solution clearly developed in the study?	1.5
4	Did the study explain recommender systems or algorithms in detail?	0.5
5	Was metrics evaluation for recommender systems explicitly used in the study?	1.5
6	Was the dataset used in the study described in detail?	0.5
7	Was the application domain introduced in the study clearly?	1
8	Was the architecture or were the parts of the suggested system described in the study?	1.5
9	Did the study provide a concise summary of its findings?	1

**Table 11 jimaging-11-00012-t011:** Future study proposals.

Potential Future Work	Studies
Enhance the offered solution.	7
Conduct more detailed reviews.	6
Include contextual information in recommendations.	7
Investigate applications in various fields.	5
Use more data or item features.	5
Test a variety of algorithms.	8
Experimentation with various hybrid recommendation models.	6
Other.	8

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
