# Peer review of "Hybrid Quality-Based Recommender Systems: A Systematic Literature Reviewâ€"

_2313-433X, 2025, doi:10.3390/jimaging11010012_

Round 1
Reviewer 1 Report
Comments and Suggestions for Authors
Some sections of the manuscript, particularly in the introduction and discussion, suffer from unclear phrasing and grammatical errors that detract from the overall readability. For example, phrases such as "technology on a global scale" and "technology impact" are vague and confusing. The writing could be polished to ensure the clarity and precision of the ideas being communicated.
The flow of ideas in certain sections, especially in the abstract and introduction, could be improved. The authors sometimes jump between concepts (e.g., from discussing the long-tail effect to hybrid algorithms) without clear transitions. A more organized structure, with better transitions between topics, would improve the paper’s overall cohesiveness.
While the paper does mention some applications of hybrid RSs in fields such as e-commerce and music, there could be more detailed examples of real-world applications, particularly from industry. Highlighting how these systems have been deployed successfully in businesses or offering more case studies could enhance the paper's relevance and appeal to a broader audience.
The paper briefly discusses the challenge of evaluating recommender systems based solely on accuracy measures. However, it would benefit from a more in-depth discussion of alternative evaluation metrics (e.g., diversity, novelty, or user satisfaction) and how they could improve the effectiveness of hybrid RSs. Including these discussions would give a more holistic view of the system performance.
Although the authors identify interesting future research areas (e.g., context-aware systems and hybridization techniques), they could provide a more detailed analysis of how these areas could be explored. Offering concrete steps for advancing these topics or identifying the tools and methodologies that might be required would strengthen the value of the discussion on future research.
Consider adding more examples or case studies from industry to demonstrate how hybrid recommender systems have been applied in practice.
Provide a more comprehensive discussion of alternative metrics used to evaluate the success of recommender systems, especially in real-world applications, and how these metrics align with user-centric objectives.
Author Response
Comments 1: Some sections of the manuscript, particularly in the introduction and discussion, suffer from unclear phrasing and grammatical errors that detract from the overall readability. For example, phrases such as "technology on a global scale" and "technology impact" are vague and confusing. The writing could be polished to ensure the clarity and precision of the ideas being communicated
Response 1:. Thank you for your constructive comments. We appreciate your comments on the clarity and grammatical accuracy of the introduction and discussion. We have carefully reviewed these sections to correct any unclear wording or grammatical errors. (see p1 to 5) (Text with red coloring)
Comments 2: The flow of ideas in certain sections, especially in the abstract and introduction, could be improved. The authors sometimes jump between concepts (e.g., from discussing the long-tail effect to hybrid algorithms) without clear transitions. A more organized structure, with better transitions between topics, would improve the paper’s overall cohesiveness.
Response 2:. Thank you for your valuable feedback regarding the flow of ideas in the abstract and introduction. We appreciate your observation about the transitions between concepts, particularly the movement from the long-tail effect to hybrid algorithms
We updated these parts to create stronger connections between subjects and increase the document's cohesion. To accomplish this, we rearranged concepts: We made sure that each notion flowed smoothly into the next by organizing them into a more logical sequence.
To help the reader follow along with the conversation and understand how one idea relates to another, we included transitional phrases and sentences.
(see p1 to 5) (Text marked in red)
Comments 3: While the paper does mention some applications of hybrid RSs in fields such as e-commerce and music, there could be more detailed examples of real-world applications, particularly from industry. Highlighting how these systems have been deployed successfully in businesses or offering more case studies could enhance the paper's relevance and appeal to a broader audience.
Response 3:. Your insightful observation regarding the necessity of providing more detailed examples of the real-world applications of hybrid recommender systems (RS) was greatly appreciated. We knew that adding particular case studies may significantly improve the document's attractiveness and applicability. Specific instances of hybrid recommender system (RS) research and application across a range of sectors are added on page 2 (Text outlined in red)
- E-commerce: (Amazon)
- Music Streaming (Spotify)
- Online Video Platforms (Netflix)
- Travel and Hospitality (Airbnb)
- Social Media (Facebook:,Linkedin)
Comments 4: The paper briefly discusses the challenge of evaluating recommender systems based solely on accuracy measures. However, it would benefit from a more in-depth discussion of alternative evaluation metrics (e.g., diversity, novelty, or user satisfaction) and how they could improve the effectiveness of hybrid RSs. Including these discussions would give a more holistic view of the system performance.
Response 4: We appreciate your constructive observations for expanding our discussion of other evaluation metrics for recommender systems. In the revised manuscript, we conducted a more thorough investigation of indicators like as diversity, uniqueness, and user happiness. We talked about how these metrics supplement accuracy measures and help to a more comprehensive evaluation of hybrid recommender systems. This update will help to demonstrate the diverse nature of system performance and provide a better understanding of how our approach may improve the user experience. See page 4-5 (Text denoted in red)
Comments 5: Although the authors identify interesting future research areas (e.g., context-aware systems and hybridization techniques), they could provide a more detailed analysis of how these areas could be explored. Offering concrete steps for advancing these topics or identifying the tools and methodologies that might be required would strengthen the value of the discussion on future research.
Response 5: We appreciate your insightful comments. We value your recommendation to thoroughly examine the highlighted future research topics, including hybridization strategies and context-aware systems. Along with a review of pertinent tools and approaches that could support this research, the new manuscript included concrete actions for furthering these issues.
we included the following specifics:
Context-Aware Systems
Hybridization
See pages 39 & 40 (Text tinted red)
Comments 6 : Consider adding more examples or case studies from industry to demonstrate how hybrid recommender systems have been applied in practice.
Response 6: Thank you for your constructive suggestion. We agree that incorporating real-world examples and case studies would greatly enhance the manuscript. In the revised version, We included several industry case studies that illustrate the practical applications of hybrid recommender systems.
- Netflix:
- Amazon:
- Spotify:
- YouTube:
See p 2 (Text with red coloring)
Comments 7 : Provide a more comprehensive discussion of alternative metrics used to evaluate the success of recommender systems, especially in real-world applications, and how these metrics align with user-centric objectives.
Response 7: Thank you for your helpful feedback. In conclusion, we recognize that assessing the performance of recommender systems necessitated a multidimensional approach that included both traditional measurements like precision and recall, as well as user-centric criteria like engagement, diversity, and serendipity. By aligning these metrics with user goals, developers may construct more effective and fulfilling recommender systems that improve the user experience and drive long-term engagement.
Several metrics were considered :
Diversity
Novelty
User satisfaction…
See pages 5 & 6 (Text in red color)

Reviewer 2 Report
Comments and Suggestions for Authors
The paper entitled "Hybrid-Based Quality Recommender Systems: A Systematic Literature Review" presents important insights into the evolution and utilization of hybrid recommender systems. However, there are some critical observations (aspects worth addressing), which underscore areas in need of enhancement and potential deficiencies.
1. First, the research questions delineated in Table 1 lack specificity. For example, RQ1, "What are the relevant studies on hybrid recommenders and what kinds of challenges can hybridization techniques typically solve?" could greatly benefit from more precise criteria regarding the specific challenges the authors are addressing. Because of this ambiguity, it becomes difficult to ascertain how each section systematically responds to these inquiries.
2. Second, while the abstract delivers a satisfactory overview of the content, the overall flow and structure could be refined. It notably lacks a succinct statement of the findings, which is vital for engaging potential readers.
3. The introduction fails to adequately articulate the scope and significance of the research. Although it provides some context, a more compelling introduction would better illustrate the importance of hybrid recommender systems in today's landscape, especially considering the notable expansion of e-commerce.
4. The paper often oscillates between specialized terminology and more accessible language, lacking a cohesive structure. For instance, terms like "quality recommender systems" and "hybrid systems" are employed; however, their definitions tend to overlap or remain ambiguous. This inconsistency, coupled with the absence of clear definitions, detracts from overall readability.
5. The section discussing challenges (such as cold start and sparsity issues) is quite informative, but it ultimately lacks depth. Although the authors offer some insights, they could enhance their analysis by delving deeper into these challenges across the various studies they examined. Moreover, it would be beneficial to include a comparison of how contemporary hybrid systems tackle these problems in contrast to traditional systems.
6. Additionally, while the authors provide an overview of the databases utilized, the clarity of the study selection and data extraction process leaves much to be desired. The inclusion and exclusion criteria (IC/EC) are referenced; however, the explanation of how these criteria were applied consistently is notably absent. For example, how did the authors ascertain relevance during the filtering process? Increased transparency regarding the methodology would greatly bolster the rigor of the review, thereby enhancing its academic value.
7. Although the paper attempts a quantitative analysis by examining publication trends, the visualizations (for instance: Fig. 12) are underutilized; this is a missed opportunity. These graphs could be supplemented with more insightful commentary. The authors should delve into why certain trends occur such as the surge in hybrid recommender research and relate these to broader industry or technological shifts.
8. The paper mentions several application areas, such as e-commerce and healthcare, however, the discussion is largely theoretical. Including more detailed case studies or examples from industry would enhance the practical value of the review. How do these systems perform in real-world settings? What are their limitations when deployed at scale?
9. The review would benefit from a more focused conclusion and a dedicated section on gaps in the literature. Although the authors do mention future research directions, these are quite generic; they could improve by identifying specific gaps that need addressing, such as challenges in scalability or integrating more advanced AI techniques into hybrid systems.
10. The section detailing the paper's results and contributions could, however, benefit from greater conciseness. The existing structure often repeats findings, failing to clearly highlight the novel insights or contributions. A more focused narrative would assist the reader in grasping the key takeaways from the review.
11. Furthermore, there are several instances in the text where the English is unclear (particularly in the methodology and results sections). Sentences tend to be overly complex or awkwardly constructed, which makes it challenging to follow the authors' arguments. A comprehensive review of the language and formatting is necessary particularly regarding proper citation and figure labelling because this would enhance the paper's overall polish and professionalism.
Comments on the Quality of English LanguageThe English could be improved
Author Response
Comments 1:First, the research questions delineated in Table 1 lack specificity. For example, RQ1, "What are the relevant studies on hybrid recommenders and what kinds of challenges can hybridization techniques typically solve?" could greatly benefit from more precise criteria regarding the specific challenges the authors are addressing. Because of this ambiguity, it becomes difficult to ascertain how each section systematically responds to these inquiries.
Response 1:Thank you for your valuable feedback regarding the specificity of the research questions in Table 1. I appreciate your insights, as they help enhance the clarity and focus of our work.
In response to your comment about RQ1, I revised the question to include more precise criteria regarding the specific challenges associated with hybrid recommender systems. This will help to clarify our focus and ensure that each section of the paper systematically addresses the outlined inquiries.
Additionally, I provided examples of the types of challenges we aim to explore within the context of hybridization techniques, which should aid in understanding the scope of our research.
RQ1 What are the relevant studies on hybrid recommenders, and how do hybridization techniques solve specific difficulties such as cold start, novelty, diversity, and user satisfaction?
See pages 5, 6,11, 31 ((Text with red coloring)
Comments 2:Second, while the abstract delivers a satisfactory overview of the content, the overall flow and structure could be refined. It notably lacks a succinct statement of the findings, which is vital for engaging potential readers.
Response 2: I am grateful for your thoughtful suggestions.This review explores recent advancements in hybrid recommender systems that combine multiple recommendation methods, addressing unique challenges and opportunities presented by big data. By examining state-of-the-art models and their real-world applications, this study identifies key gaps and directions for future research. It also employs ASReview, an active-learning tool, to streamline the literature selection process efficiently
See abstract page 1
Comments 3 The introduction fails to adequately articulate the scope and significance of the research. Although it provides some context, a more compelling introduction would better illustrate the importance of hybrid recommender systems in today's landscape, especially considering the notable expansion of e-commerce.
Response 3: Thank you for this insightful feedback. The research has been revised to emphasize the importance of hybrid recommender systems in today's data-driven environment, especially in the e-commerce sector. The revised introduction highlights the transformative impact of hybrid systems on user experience, business outcomes, and competitive advantage. It aims to provide a clearer understanding of the research's relevance and contributions to advancing hybrid recommender system applications in various sectors. The revised text is marked in red. (see pages 2 to 5) (Text marked in red)
Comments 4 :The paper often oscillates between specialized terminology and more accessible language, lacking a cohesive structure. For instance, terms like "quality recommender systems" and "hybrid systems" are employed; however, their definitions tend to overlap or remain ambiguous. This inconsistency, coupled with the absence of clear definitions, detracts from overall readability.
Response 4:Thank you for your valuable feedback regarding the terminology and structure of the paper.
The term "quality" in the context of “Hybrid Quality-Based Recommender Systems” typically refers to the system's ability to deliver recommendations that are more accurate, relevant, and satisfying for the user. Quality metrics may include precision, relevance, diversity, novelty, and user satisfaction.
However, in the case of "hybrid systems," we have omitted the term "recommender," which specifically refers to "hybrid recommender systems."
See pages 4, 5 (text in red color)
Comments 5: The section discussing challenges (such as cold start and sparsity issues) is quite informative, but it ultimately lacks depth. Although the authors offer some insights, they could enhance their analysis by delving deeper into these challenges across the various studies they examined. Moreover, it would be beneficial to include a comparison of how contemporary hybrid systems tackle these problems in contrast to traditional systems.
Response 5: I appreciate your detailed observations. To address the cold start problem in content-based recommender systems, businesses can establish user profiles using demographic attributes and seek explicit feedback from new users. Techniques like matrix factorization and singular value decomposition can simplify complex rating matrices. To address data sparsity, a hybrid method combining Collaborative Filtering with Sequential Pattern Analysis and contrastive learning can improve model performance, boosting accuracy up to 88.70% with optimized hyperparameters.
See pages 29 to 33 (Text rendered in red)
Comments 6: Additionally, while the authors provide an overview of the databases utilized, the clarity of the study selection and data extraction process leaves much to be desired. The inclusion and exclusion criteria (IC/EC) are referenced; however, the explanation of how these criteria were applied consistently is notably absent. For example, how did the authors ascertain relevance during the filtering process? Increased transparency regarding the methodology would greatly bolster the rigor of the review, thereby enhancing its academic value.
Response 6: Thank you for your useful advice. The study used a rigorous methodology to select studies, identifying 5,857 primary studies from five digital libraries. Scopus and Springer were used, with priority given to articles published between 2020 and 2024. After a coarse selection phase and systematic review, 52 high-quality studies were chosen, enhancing the review's transparency and academic value.
See pages 30,31 ((Text tinted red)
Comments 7: Although the paper attempts a quantitative analysis by examining publication trends, the visualizations (for instance: Fig. 12) are underutilized; this is a missed opportunity. These graphs could be supplemented with more insightful commentary. The authors should delve into why certain trends occur such as the surge in hybrid recommender research and relate these to broader industry or technological shifts.
Response 7: I am thankful for your critical analysis. The rapid increase in publications on hybrid recommender systems can be attributed to the emergence of deep learning techniques and the adoption of cloud computing, which facilitate the management of complex data. This trend reflects a growing academic interest driven by the industry's demand for personalized recommendation engines, further accelerated by the introduction of transformer models. Although 2024 shows a temporary dip in publication numbers due to early-year data collection, this does not indicate a long-term decline. Overall, the findings suggest a dynamic and evolving field that merits close monitoring as research progresses
See page 23 -24 (Text marked in red)
Comments 8: The paper mentions several application areas, such as e-commerce and healthcare, however, the discussion is largely theoretical. Including more detailed case studies or examples from industry would enhance the practical value of the review. How do these systems perform in real-world settings? What are their limitations when deployed at scale?
Response 8: Thank you for your thoughtful feedback. High computing needs and latency concerns caused by sophisticated models make it difficult to deploy hybrid recommender systems at scale. Integrating various data sources, retraining on a regular basis, and assuring interpretability all bring additional hurdles. Cold start issues, data scarcity, and algorithm scalability are all factors that affect performance. It remains difficult to balance real-time customization with system response time and expenses.
see pages 2, 3, 33 (Text with red coloring)
Comments 9: The review would benefit from a more focused conclusion and a dedicated section on gaps in the literature. Although the authors do mention future research directions, these are quite generic; they could improve by identifying specific gaps that need addressing, such as challenges in scalability or integrating more advanced AI techniques into hybrid systems.
Response 9: Thank you for your constructive feedback regarding the need for a more focused conclusion and a dedicated section on gaps in the literature related to hybrid recommender systems. We appreciated your insights and agreed that highlighting specific research gaps would enhance the review's clarity and relevance.
In the revised conclusion, we summarized the key findings of the review while emphasizing the importance of addressing specific challenges in the field of hybrid recommender systems. We stressed that the current landscape was evolving rapidly due to technological advancements, yet several critical areas remained underexplored.
See p 39, 40 (Text accented in red)
Comments 10: The section detailing the paper's results and contributions could, however, benefit from greater conciseness. The existing structure often repeats findings, failing to clearly highlight the novel insights or contributions. A more focused narrative would assist the reader in grasping the key takeaways from the review.
Response 10: The discussion on hybrid recommender systems was enhanced by incorporating practical examples from e-commerce platforms like Amazon and Netflix. These platforms used hybrid approaches, combining collaborative filtering and content-based methods to deliver personalized experiences. In healthcare, hybrid systems supported personalized treatment plans by integrating patient history with clinical research data. IBM Watson Health utilized hybrid systems to assist clinicians in making data-driven treatment decisions. The paper also refined the narrative to emphasize novel insights and key takeaways, providing a balanced overview of hybrid recommender systems' capabilities and limitations in practical scenarios.See pages 20 to 24 ( Text highlighted in red)
Comments 11: Furthermore, there are several instances in the text where the English is unclear (particularly in the methodology and results sections). Sentences tend to be overly complex or awkwardly constructed, which makes it challenging to follow the authors' arguments. A comprehensive review of the language and formatting is necessary particularly regarding proper citation and figure labelling because this would enhance the paper's overall polish and professionalism.
Response 11: Thank you for your valuable feedback. We appreciate your insights regarding the clarity of our writing, particularly in the methodology and results sections. We conducted a comprehensive review of the language to simplify overly complex sentences and improve the overall flow of our arguments. Additionally, we ensured that citations and figure labels were formatted correctly to enhance the paper's professionalism and readability.

Round 2
Reviewer 1 Report
Comments and Suggestions for Authors
Ok
Reviewer 2 Report
Comments and Suggestions for Authors
Accept